



# A new base of wind turbine noise measurement data and its application for a systematic validation of sound propagation models

Susanne Könecke[1,2], Jasmin Hörmeyer[3], Tobias Bohne[1], and Raimund Rolfes[1]

[1]Leibniz University Hannover, Institute of Structural Analysis, Appelstraße 9A, 30167 Hannover, Germany
[2]Previous research was published under the name Susanne Martens
[3]GICON - Großmann Ingenieur Consult GmbH, Tiergartenstraße 48, 01219 Dresden, Germany

**Correspondence:** Susanne Könecke (s.koenecke@isd.uni-hannover.de)

**Abstract.** Extensive measurements in the area of wind turbines were performed in order to validate a sound propagation model which is based on the Crank-Nicolson Parabolic Equations method. The measurements were carried out over a flat grass-covered landscape and under various environmental conditions. During the measurements, meteorological and wind turbine performance data were acquired and acoustical data sets were recorded in distances of 178, 535, and 845 m to the wind turbine. By processing and analyzing the measurement data, validation cases and input parameters for the propagation model were derived. The validation includes five groups that are characterized by different propagation directions, i.e. down-, cross- and upwind conditions in varying strength. Comparing measured and modeled propagation losses, a general good agreement is observed for all groups. Considering all groups and distances, the absolute averaged difference of the measured and modeled losses in total sound pressure level is 0.9 dB. At large distances, the propagation losses are slightly underestimated by the model. The model represents the measured propagation losses in the 1/3 octave spectra well. As in the measurements, frequency-dependent maxima and minima are identified and losses generally increase with increasing distance and frequency. For good results in upwind situations, turbulence has been considered in the model. The data sets used in the validation are available for further research.

## 1 Introduction

From 2009 to 2020, the global capacity of onshore wind turbines has increased from 157 to 707 GW. A further growth of 399 GW is expected in the years 2021 to 2025 (Lee and Zhao, 2021). With the expansion of wind energy and the decreased distance between turbines and local residents, the noise emission from wind turbines and its propagation has come into focus. This paper addresses the latter issue - the sound propagation of wind turbines.

Various analytical and numerical modeling techniques have been applied to predict the propagation of wind turbine noise (Bérengier et al., 2003). The most well known physically based sound propagation models are the ray tracing and the parabolic equations (PE) method. With a focus on low-level sources, most of the models are verified using the benchmark of Attenborough et al. (1995) and are partially validated with measured data. Performing and evaluating acoustic measurements requires a lot of effort and cost, especially for high-level sources such as wind turbines. As a result, numerical models for high-level





sources have often been verified by analytical results (e.g. Lee et al. (2016); Cotté (2019)) or engineering models (e.g. Bolin
and Boué (2009); Kaliski and Wilson (2011)) in the past, but are less validated by measured data.

In addition to analytical solutions, Lee et al. (2016) compared numerical results with far-field acoustic measurements to
validate the PE method. For this, two loudspeakers were placed at 20 and 80 m height of a meteorological mast and seven
microphones were positioned at 2 m height and in 500 to 1700 m distance from the mast. Single tone frequencies (125, 250,
500, and 1000 Hz) were used as sources. In downwind direction, a good agreement between measurements and model was
observed for both speaker heights. Since no turbulence was taken into account in the model, the sound level in upwind direction
was strongly underestimated by the PE method.

Shen et al. (2005) carried out measurements with a loudspeaker including the detection of fluid and acoustic quantities to
validate four different propagation models, namely the PE-based WindStar (Barlas et al., 2017a) and the ray-tracing based
Nord2000 (Plovsing, 2014), as well as the ISO 9613-2 (1996) and DK-BEK513 (2019) standards. The loudspeaker was placed
at a height of 109 m, the atmospheric conditions were recorded by a meteorological mast and acoustic measurements were
performed with 11 microphones placed at different distances to the turbine. White noise and band-limited white noise were
applied as signals. For two different wind shears, the measured and modeled 1/3 octave spectra (125 to 1000 Hz) were compared
and the average difference of the total sound pressure level was determined. For all considered models a good agreement with
the measurements was achieved. Depending on the microphone position and propagation model, the difference of measured
and modeled data were between -3.43 and 2.45 dB. The comparison was performed for crosswind conditions.

In Prospathopoulos and Voutsinas (2005), a sound propagation model based on ray-traying was validated by acoustic mea-
surements in the area of one wind turbine. For this purpose, the sound pressure level was recorded somewhere between 70
and 88 m and in 530 m distance to the turbine. Validating the model, the measured and modeled propagation losses between
the 70/88 and 530 m were compared in 1/3 octave bands. Using one scenario as an example, i. e. for downwind conditions
and flat land, a good agreement was shown. However, as the focus of the paper was the investigation of different propagation
effects, no further validation cases were presented. Moreover, with a hub height of 60 m, the investigated wind turbine does
not correspond to the current scales of up to 165 m.

As part of the project 'Noise and energy optimisation of wind farms', extensive measurements were carried out to validate
the Nord2000 propagation model for the use of wind turbine noise (Søndergaard and Plovsing, 2009). This ray-tracing based
model was validated by several field measurements with different sources - namely with two loudspeakers, a single wind turbine
and a whole wind farm. Data from a 100m measurement mast were used to determine sound speed profiles. In general, good
agreements were obtained for simple and also complex conditions regarding meteorology and landscape. For the loudspeaker
test in flat terrain, distances of up to 1500 m were considered. Showing an average deviation of 0.1 dB, very good results
were achieved in downwind direction. With an average deviation of 4.3 dB, higher differences of measured and modeled data
were examined in upwind direction. Herein, the predicted propagation losses were 4 dB lower than the measured ones. Note
that turbulence constants were taken into account in the model. For the validation with a single wind turbine, only downwind
conditions were considered. The total levels were within 1 dB on average, but the measured and predicted 1/3 octave spectra
differed to some extent.





For various reasons, the data provided by the literature are not suitable for validating sound propagation models applied to wind turbines. Firstly, loudspeakers do not reflect the spatial and time-dependent sound characteristics of a wind turbine. Second, although atmospheric conditions are often measured, the specific measured values are not available to the reader. Consequently, some of the input parameters for a sound propagation model cannot be derived and the findings cannot be used to validate further models. In order to validate and to improve sound propagation models for wind turbines, open-source measurement data is helpful. For this purpose, a detailed presentation, processing and analysis of the acoustic and meteorological measurement data as well as a subsequent data publication is necessary. For this reason, the objectives of this paper are

(1) to introduce comprehensive measurements of acoustic and atmospheric quantities close to a real Multi-MW-turbine

(2) to prepare, combine and analyze acoustic, atmospheric and wind turbine measurement data for the validation of sound propagation models

(3) to apply prepared data-sets for the systematic validation of a numerical sound propagation model based on the PE method taking into account different propagation directions

(4) to provide validation data of wind turbine sound propagation for further research purposes.

The paper is structured as follows. In Sec. 2, the methodology is described in detail. The measurements, the PE-based sound propagation model applied and the modeling of a representative sound source of a wind turbine are presented. The focus of the section is in particular on the derivation of input parameters from measurement data for the model. All results are given in Sec. 3. The acoustic and atmospheric measurement data are analyzed and the measurement- and model-based propagation losses are compared using 1/3 octave bands and total sound pressure levels. Subsequently, the results are discussed considering model assumptions and the impact of input parameters.

## 2 Methodology

In this section, an overview of the modeling approach is given first. Then the focus is on the measurements, they are presented in detail and the derivation of input parameters is discussed extensively. Finally, the validation process is described.

### 2.1 Modeling

In this work, extensive wind turbine noise measurement data are used to validate a propagation model with a simplified wind turbine sound source. In general, the sound propagation is essentially determined by the geometric attenuation. In addition, sound propagation is influenced by the ground and by atmospheric aspects such as air absorption, refraction and scattering. According to Salomons (2001), the sound pressure level $L_p$ at the place of immission can be calculated as a function of the frequency $f$:

$$L_p(f) = L_W(f) - \underbrace{10 \cdot \log 4\pi R^2 - \alpha_L(f) \cdot R/1000}_{Attenuation\ terms} + \Delta L(f) \tag{1}$$

with the sound power level of the wind turbine $L_W$, the distance $R$, the atmospheric air coefficient $\alpha_L$ and the term $\Delta L$, which describes additional attenuation due to further propagation effects. In Eq. 1 attenuation terms are subtracted from the





sound power level of the wind turbine. The attenuation terms include the geometrical spreading (first term) and air absorption (second term) which are both dependent on the distance $R$. Furthermore, the air absorption depends also on the atmospheric coefficient $\alpha_L$ which is calculated as a function of frequency, temperature and humidity according to Bass et al. (1995). The last attenuation term $\Delta L$ describes the sound propagation loss due to ground effects as well as atmospheric refraction and scattering. In this work, $\Delta L$ is determined using the Crank-Nicolson Parabolic-Equation (CNPE) method.

The CNPE method is an efficient methodology for the calculation of sound propagation over large distances, because backscattering is neglected and the calculations are only performed in the propagation direction (Salomons, 2001). As a result, it is a common approach for predicting the propagation of wind turbine noise (Lee et al., 2016; Barlas et al., 2017a, b; Zhu et al., 2018; Cotté, 2019). The propagation model of this work follows the descriptions in West et al. (1992) and in Salomons (2001) and is shortly introduced in the following. Therefore, the CNPE method is simplified into a two-dimensional form on

the basis of an axisymmetric approximation. The two-dimensional Helmholtz equation is given as

$$\frac{\partial^2 q}{\partial r^2} + \frac{\partial^2 q}{\partial z^2} + k_{\text{eff}}^2 \cdot q = 0 \tag{2}$$

where the sound field $q$ is dependent on the cylindrical coordinates $r$ and $z$ and is associated with the complex pressure amplitude $p$ by $q = p\sqrt{r}$. Moreover, the local effective wavenumber $k_{\text{eff}} = \omega/c_{\text{eff}}$ with the angular frequency $\omega$ and the effective sound speed $c_{\text{eff}}$ is considered in Eq. 2. Calculating the sound pressure field $q$, a wide-angle parabolic equation is solved

using the Crank-Nicolson method in r-direction and central finite differences in z-direction. In the simulations of this work, a discretization equal to one-tenth of the wavelength $\lambda$ is chosen in both vertical and horizontal direction (i.e. $\Delta r = \Delta z = \lambda/10$) providing sufficient accuracy. To simulate free field conditions in z-direction, a perfectly matched layer is used at the upper boundary of the computational domain. The lower boundary is defined by the acoustic ground impedance. For the characterization of those ground impedances, the Delany-Bazley-Miki model (Miki, 1990) is used accounting for specific

ground properties. This model is based on an empirical ground model by Delany and Bazley (1970). Since the site of the measurements was predominantly grass-covered, a flow resistance of 200 $\text{kPa s/m}^2$ is chosen for the Delany-Bazley-Miki model. This, according to various publications, is a typical value for grassland. Moreover, in view of the measurements a flat terrain is assumed.

The present CNPE model uses a second-order starting field described in Salomons (2001) by

$$q_0 = \sqrt{ik_a}(A_0 + A_2 k_a^2 z^2) exp\left(-\frac{k_a^2 z^2}{B}\right) \tag{3}$$

where the variables $A_0$, $A_2$ and $B$ are obtained from empirical studies. Generally, the starting field represents a monopole sound source.

However, to represent a wind turbine as a source, the approach from Barlas et al. (2017b) is adopted. In this approach the wind turbine is reduced to three incoherent point sound sources, which are located at the rotor blade tips, more precisely at 85%

of the rotor length, in the three-dimensional field. According to Oerlemans et al. (2007), the sound radiation of wind turbines is dominant at this position. Transferred to a two-dimensional field, the point sound sources are located at hub height $h$ and at



$\pm 85\%$ of the rotor length $l$:

$$h_s = h \pm 0.85l. \tag{4}$$

For this simplified representation of a wind turbine, one simulation is performed for each sound source. Subsequently, the
simulation results are logarithmically summed (Barlas et al., 2017b). In this paper, measurement data averaged over 10 minutes
are used, so that the sources are assumed to be steady. A short comparison with a monopole source is presented in Sec. 3.3.
Moreover, in the present work atmospheric turbulence is taken into account in certain wind directions, more precisely in strong
upwind directions. In the upwind direction, the sound is refracted upwards, resulting in shadow zones where sound waves can-
not enter directly. However in reality, the sound waves are scattered into the shadow zone by turbulent eddies. Consequently,
the consideration of turbulence is necessary in upwind direction. In this work, the turbulent fields are characterized and imple-
mented according to Salomons (2001). Therefore, turbulence is described by random fields of fluctuating values in atmospheric
sound propagation. To characterize the random fields, the Gaussian correlation function

$$B(R) = \mu^2 e^{(-R^2/a^2)} \tag{5}$$

with

$$\mu^2 = \frac{\sigma_T^2}{4T_0^2} + \frac{\sigma_W^2}{c_0^2} \tag{6}$$

is used in this work. Herein, the turbulent field is described in the two-dimensional vector R and the correlation length (a)
is set to 1 m according to Salomons (2001). The turbulent fluctuation $\mu$ from Eq. 5 is characterized with the variance of the
temperature $\sigma_T^2$ and the wind speed $\sigma_W^2$, related to the values of the temperature $T_0$ and the sound speed $c_0$ near the ground
(see Eq. 6). Thus, the turbulent fluctuation and therefore the turbulent fields can be determined on the basis of measured
data. In the propogation model, the turbulence is implemented by multiplying the sound field at the position $r$ by a phase
factor after each calculation step. The characteristic fields are assumed to be 'frozen' realizations of the turbulent atmosphere
and are determined by a random number generator. In order to obtain sufficiently accurate predictions of the sound pressure
levels, several realizations must be carried out and subsequently the calculated levels must be logarithmically averaged. In
this work, 50 realizations are carried out for the cases with strong upwind conditions. Since the consideration of turbulence is
computationally intensive and is only decisive for upwind conditions, turbulence is neglected in the other wind directions.

For further information about the model and the implementation of turbulence Salomons (2001) is recommended. The
limitation due to the source approximation and further constraints of the model as well as the influence of ground conditions
and turbulence on the sound propagation are discussed in section 3.3.

## 2.2 Measurements

The data sets selected for the validation originate from one of five measurement campaigns, which are described in detail
in Martens et al. (2020). In this work, only a very brief overview is given in order to provide the reader a rough understanding
of the measurements. The measurement data used originates from a measurement campaign performed close to a turbine in





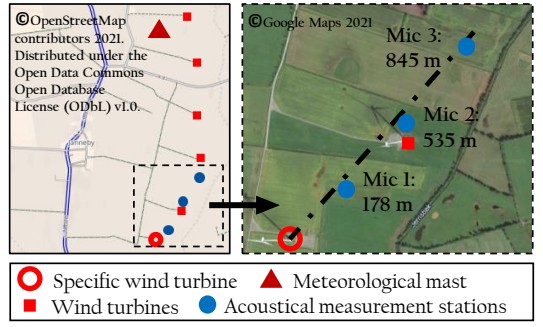

**Figure 1.** Overview map of the wind farm and detailed measurement plan including the position of specific wind turbine, acoustical measurement stations and meteorological mast

a wind farm in northern Germany. The landscape of the measurement site is characterized as flat and homogeneous and is predominantly covered with grass. Acoustic and meteorological data as well as turbine-specific parameters of the wind farm,

i.e. Supervisory Control And Data Acquisition (SCADA) Data, were recorded synchronously over a period of several months.

The measurement environment as well as the position of the measurement instruments and turbines are shown in Fig. 1. The distances from the wind turbine to the acoustic stations are summarized in Tab. 1. During the campaign, three acoustic measurement stations recorded sound pressure levels, 1/3 octave bands, and audio at a sampling rate of 51 kHz. To avoid additional extraneous noise from natural sources, the acoustic measurement devices were positioned at least 10 m from possible

disturbances. A challenging task for acoustic measurements in the free field is the reduction of wind-induced noise at the microphone. These noises can strongly distort measurement data, especially in the low-frequency range. Using a combination of a nose cone, a 90 mm standard windscreen and an in-house developed 220 mm secondary windscreen, the wind-induced noise at the microphones was effectively reduced during the measurements. The development and further investigations concerning the in-house developed windscreens are described in Martens et al. (2019). An example of an acoustic measurement station

with windscreens is shown in Fig. 2. Generally, the height of each sound level meter was 1.70 m. Moreover, the systems were powered by solar panels and an additional external battery during the time of measurements.

Synchronously to the acoustic recordings, extensive meteorological measurements were performed describing the lower atmosphere. A 100 m high measuring mast is permanently positioned in the wind farm, which records temperature and humidity as well as wind speed and wind direction at different heights having a resolution of ten minutes. The data of wind speed is also

available in 1 Hz, which is important for the determination of the sound speed profile. The corresponding measurement setup is illustrated in Fig. 2 as well as the position of the measurement mast is given in Fig. 1. According to this, the meteorological mast is located in the center of the wind farm. In addition to acoustic and meteorological parameters, the operational data of all wind turbines in the wind farm were detected, such as rotor speed and electrical power. Herein, the data is provided at a resolution of ten minutes.





**Table 1.** Horizontal distances from wind turbine to acoustic measurement stations

| Microphone | Distance to wind turbine |
| --- | --- |
| Mic 1 | 178 m |
| Mic 2 | 535 m |
| Mic 3 | 845 m |

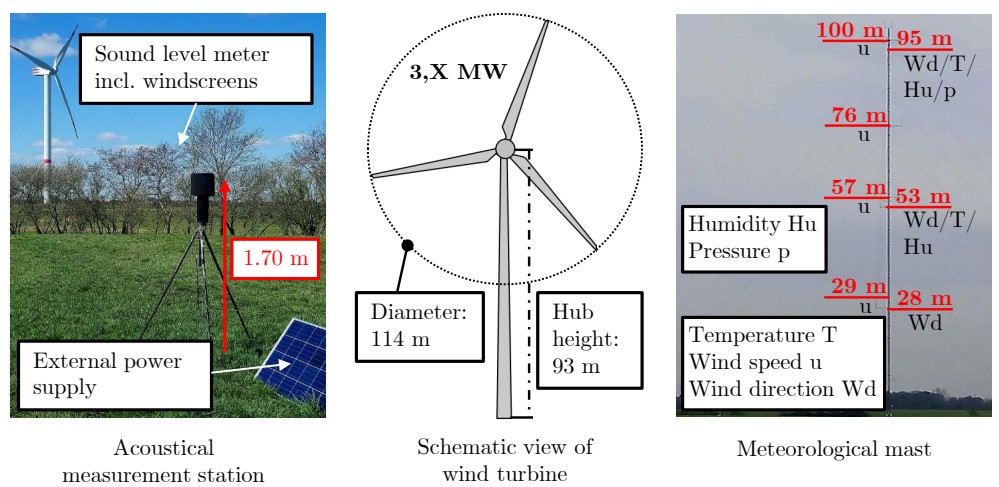

**Figure 2.** Overview of measurements systems and wind turbine

## 2.3 Determination of input parameters

As mentioned above, sound propagation in the atmosphere is influenced by air absorption, turbulence and atmospheric refraction. The parameters for estimating the air absorption are derived directly from the measurements. Temperature and humidity at 53 m are used to calculate the atmospheric coefficient $\alpha_L$ and hence, the air absorption using the approach of Bass et al. (1995). The parameters characterizing the turbulences are also determined on the basis of the measurements. Having the standard deviation of measured temperature and wind speed data, the variance and consequently the turbulent fluctuation in Eq. 6 is calculated. Herein, the temperature at the bottom of the tower is used as the reference $T_0$ and the sound speed is defined as

$$c_0 = \sqrt{\kappa R T_0} = 20.05 \sqrt{T_0}, \tag{7}$$

with the specific heat capacity $\kappa$ set to 1.4 and the gas constant $R$ of dry air to 287 Jkg$^{-1}$K$^{-1}$.

Regarding the atmospheric refraction, the vertical profile of the sound speed is essential. Generally, the sound speed $c$ at height $z$ is calculated with Eq. 7. However, in the moving atmosphere, the speed of sound is superimposed with the prevailing wind speed and is obtained in the direction of propagation with

$$c_{\text{eff}} = c_0 - |u| \cdot \cos(\gamma). \tag{8}$$



**Table 2.** Criteria for stability classes in dependence of the wind shear exponent $\alpha$ according to van den Berg (2008)

| stability class | wind shear exponent $\alpha$ |
| --- | --- |
| (moderately–very) stable | $\alpha > 0.4$ |
| slightly stable | $0.2 < \alpha < 0.4$ |
| neutral | $0.1 < \alpha < 0.2$ |
| (very–slightly) Unstable | $\alpha < 0.1$ |

Here, the second term describes the wind component in sound propagation direction, which is determined using the wind speed $u$ and the angle $\gamma$ between the wind direction and the sound propagation direction. Consequently, the direction of sound propagation corresponds to the angular relationship between the turbine and the microphones. Eqns. 7 and 8 are used to calculate the effective sound speed $c_{\mathrm{eff}}$ at the measurement heights illustrated in Fig. 2, i.e. at 29, 57, 76 and 100 m. On the basis of the measured data, it was verified that the wind direction does not change significantly with height.

However, a high discretization of the $c_{\mathrm{eff}}$-profile is required for the CNPE method. To determine $c_{\mathrm{eff}}$ from the ground to the maximum height of the computational domain, the log-linear approach introduced in Heimann and Salomons (2004) is followed. Accordingly, at the height $z$, $c_{\mathrm{eff}}$ can be described as a function of the coefficients $a_0$, $a_{\mathrm{log}}$ and $a_{\mathrm{lin}}$ and the roughness length $z_0$ using

$$c_{\mathrm{eff}} = a_0 + a_{\mathrm{log}} \ln \frac{z + z_0}{z_0} + a_{\mathrm{lin}} z. \qquad (9)$$

A value of 0.05 m is chosen as the roughness length, which is considered to be representative for the site according to available turbine reports. The coefficient $a_0$ corresponds to the speed of sound and is calculated via Eq. 7 using the measured temperature at a height of 53 m. The logarithmic and linear coefficients $a_{\mathrm{log}}$ and $a_{\mathrm{lin}}$ are determined using a pseudoinverse (see Golan (1995) for foundations). The accuracy of the fitted vertical profile of $c_{\mathrm{eff}}$ is given by the Root Mean Square Error (RMSE) values. A comparison between the original value of $c_{\mathrm{eff}}$ at sensor heights and the estimated vertical profile of the sound speed is shown in Sec. 3.1.

Lately, information on atmospheric stability is needed to classify different propagation conditions and thus to derive diverse validation cases. In this work, the stability conditions are described on the basis of the dimensionless wind shear exponent $\alpha$. For each ten minutes averaging period, $\alpha$ is determined by the power law expression:

$$\frac{u_{z_2}}{u_{z_1}} = \left(\frac{z_2}{z_1}\right)^{\alpha}, \qquad (10)$$

where the mean horizontal wind speeds $u$ in m/s at the heights $z_1$ and $z_2$ are applied. Calculating $\alpha$, the measured wind speed at $z_1 = 29$ m und $z_2 = 100$ m are used. Subsequently, the stability is divided into the five classes which are specified in many publications, e.g. van den Berg (2008), and are listed in the Tab. 2.



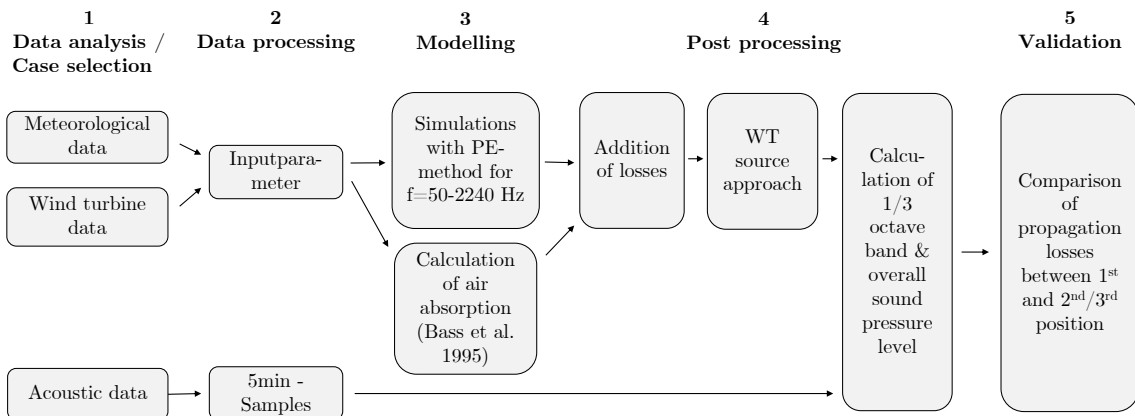

**Figure 3.** Scheme for validating a sound propagation model with wind turbine noise measurement data

## 2.4 Validation procedure

The scheme for validating the sound propagation model with the measurement data is shown in Fig. 3. The procedure is divided into five steps.

In the first step, the validation cases are selected based on a data analysis. When selecting the validation cases, it was
guaranteed, that homogeneous atmospheric conditions are present. For this purpose, the approach described in Argyle and Watson (2014) was adopted. Accordingly, the atmospheric data sets used were compared with data sets recorded within 20 minutes before and after. The atmosphere was classified as inhomogeneous if in this period of 50 minutes the wind speed varied by more than 20%, wind direction by 15° or temperature by 0.5°C. In addition to that approach, it is assumed that no particular atmospheric phenomena prevailed during the recording of the selected data sets. For example, it was ensured that no
low-level jets were present. Thus, the measured wind speed increases steadily with height and no local wind maxima are shown in the vertical profile. Besides homogeneous conditions, it was ensured that the data did not deviate from the power curve of the turbine and that the noise of the wind turbine was dominant.

In the second step of the validation, the measured data is processed. Based on the meteorological and wind turbine data the input parameters for the sound propagation model are derived. Hub height and rotor length of the wind turbine are considered
for the representation of the source. Moreover, the receiver height in the model is set equal to the microphone height during the measurements. Consequently, the same relative receiver positions to the wind turbine are examined in measurements and simulations. In addition to the measurement geometry and the wind turbine related parameters, the calculated profile of the effective sound speed, which occurred during the measurement time, is implemented (see Sec. 2.1). Acoustic data samples of five minutes are processed for the comparison of modeled data. In order to guarantee a high level of wind turbine noise, the
acoustic data sets were analyzed and checked by listening tests (see Sec. 3.1).

In the third step, simulations are performed for 1/3 octave bands with center frequencies ($f_i$) from 80 to 2000 Hz. Higher 1/3 octave bands have been neglected because of the typical emission spectra of wind turbines and especially because of the





atmospheric absorption. Due to measurement inaccuracies caused by wind-induced noise at the microphone, bands lower than 80 Hz are also excluded. Within the 1/3 octave bands, a sampling rate of $\Delta f = 5$ Hz is chosen, which proved to give sufficient accuracy. The simulations with the PE method are performed for point sound sources at three wind turbine related heights, i.e. at hub height $h$ and at $\pm 85\%$ of the rotor length $l$. In addition, the air absorption is calculated for the same frequency range.

In the post processing of the modeled data (step 4), the sound pressure level per frequency is calculated according to Eq. 1. Herein, the sound power level is set to 0, so that in this work only the propagation losses are taken into account. Moreover, the wind turbine source approach is applied to the calculated relative sound pressure level. Between lower and upper limit frequencies of the 1/3 octave bands, the modeled results at the receiver location $m$ are summed logarithmically:

$$L_{p,i}(f) = 10 \cdot \log_{10}(10^{\frac{L_{p,1}}{10}} + 10^{\frac{L_{p,2}}{10}} + ... + 10^{\frac{L_{p,n}}{10}}) \tag{11}$$

where $L_{p,i}(f)$ is the calculated relative sound pressure level with the bandnumber i. For the same frequency range, the measured data is also processed to 1/3 octave bands averaged over five minutes with dominant wind turbine noise. Besides 1/3 octave bands, the total sound pressure level between 80 to 2000 Hz is determined analogously for measured and modeled data.

In the last step, a comparison of measured and simulated results is performed. Since the sound source cannot be accurately reproduced in either the simulation or the measurements, the first receiver is used as a reference to calculate the propagation loss in both cases. In this way, additional error impacts due to inaccurate representation of the wind turbine can be reduced. The propagation loss is therefore estimated between the first microphone and other microphone positions by

$$\Delta L_p = L_{p,1} - L_{p,m}, \tag{12}$$

where $L_{p,m}$ is the relative sound pressure level at the receiver position $m$. Hence, $L_{p,1}$ is the relative sound pressure level at the first microphone position.

## 3 Results and discussion

For the comparison of measured and simulated data, different validation cases were derived. In all cases, the wind turbine is characterized by a hub height of 119 m and a rotor diameter of 114 m. The receiver positions are at a height of 1.70 m and at distances of 178, 535 and 845 m to the turbine. The selected validation cases are summarized in Tab. 3 and are grouped in terms of the propagation direction. They are grouped into light downwind (case 1, 2), cross/downwind (case 3, 4), cross/upwind (case 5, 6), light upwind (case 7, 8) and strong upwind (case 9, 10). In the figures of the paper, the different groups are color-coded in red (downwind, Case 1) through orange and green to blue (upwind, Case 10). Each group contains two validation cases that have very similar propagation characteristics. The data acquisition within a group took place in the same night and within 2 hours. With strong upwind, Cases 9 and 10 represent a special group, as they are complex and difficult to model due to the consideration for turbulence.

In this section the different validation cases are first analyzed in terms of environmental conditions and acoustic properties. Subsequently, the validation is performed by comparing measured and modeled propagation losses per 1/3 octave band and





**Table 3.** Overview of validation cases

| Case | Temperature at 53 m | Humidity at 53 m | Propagation-direction | Stability of atmosphere |
|---|---|---|---|---|
| 1 | 5.6 °C | 77% | light downwind | slightly stable |
| 2 | 6 °C | 72% | light downwind | slightly stable |
| 3 | 8.9 °C | 53% | cross/downwind | moderately stable |
| 4 | 9 °C | 53% | cross/downwind | moderately stable |
| 5 | 12 °C | 48% | cross/upwind | slightly stable |
| 6 | 13.1 °C | 47% | cross/upwind | slightly stable |
| 7 | 14 °C | 57% | light upwind | very stable |
| 8 | 13.8 °C | 58% | light upwind | very stable |
| 9 | 14.6 °C | 75% | strong upwind | moderately stable |
| 10 | 14.5 °C | 77% | strong upwind | moderately stable |

total sound pressure levels. Afterwards, the validation results are discussed regarding model assumptions and the effect of input
parameters.

### 3.1 Analysis of measured data

#### 3.1.1 Environmental conditions

Since wind direction and wind speed are key determinants of the sound speed profile, these parameters are particularly important for sound propagation and are described using Fig. 4. As a representative of other heights, the measured wind speed and wind direction at 57 respective 50 m at the time of validation cases are shown in Fig. 4(a). In Fig. 4(b), the calculated profiles and the measured values of the effective sound speed are also given. The different groups of propagation direction are clearly seen. The cases of each group provide similar characteristics regarding wind direction, wind speed and effective sound speed profile. Since the data was recorded within the same hour, the cases in strong upwind direction show almost the same effective sound speed profile. The accuracy of the calculated sound speed profiles is assessed by the Root Mean Square Error (RMSE). This value indicates the average deviation of the profile from the measured values. The RMSE values were calculated for the fitting of sound speeds over all cases. In Fig. 5, the fitting of four cases is shown as an example. Herein, the best, the worst and two average fittings are considered. The averaged RMSE value of all cases is $0.04\,\mathrm{m/s}$, so that in general a very good fitting of the sound speed profile is concluded. This assessment is based on a comparison with literature values, where values of about $0.15\,\mathrm{m/s}$ are described as sufficiently accurate (Heimann and Salomons, 2004). The worst fit is seen for case 9 with a RMSE of $0.14\,\mathrm{m/s}$. The best fit is achieved for case 4 with RMSE=$0.0017\,\mathrm{m/s}$.

The profile of the effective sound speed generally displays the different atmospheric stabilities. However, a better measure for the stability is the wind shear exponent which, calculated according to Eq. 10, is shown for all cases in Fig. 6. With an





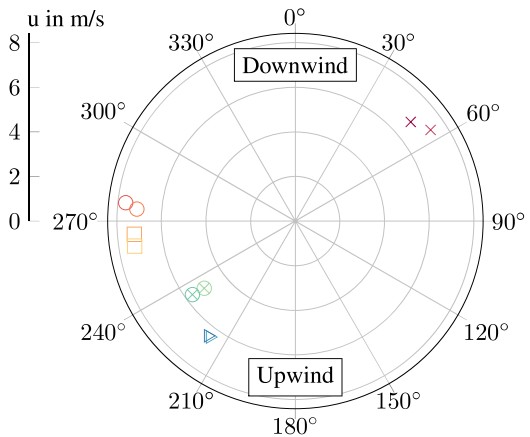

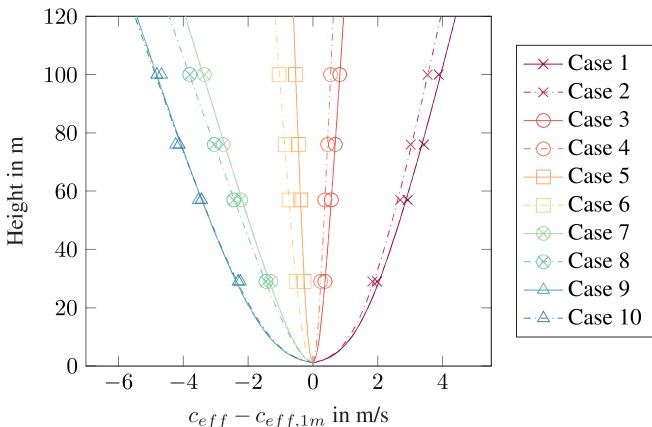

(a) Distribution of measured wind speed and wind direction at 57 respective 50 m height

(b) Derived profiles of effective sound speed (lines) based on measured data at four heights (markers), normalized with the effective sound speed at 1 m

**Figure 4.** Wind rose and effective sound speed profiles at times of the validation cases. The wind directions are related to the microphone positions, so that a direction of 0° indicates downwind conditions whereas a direction of 180° represents upwind conditions. for both graphs the legend at the right side is used.

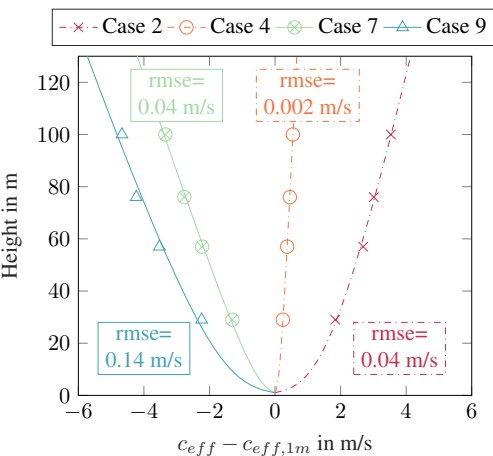

**Figure 5.** Normalized effective sound speed including the root mean square error for the best (0.002 m/s) and worst (0.14 m/s) approximation as well as two approximations close to the average RMSE (0.04 m/s)

exponent of over 0.6, cases are assigned to a very stable atmosphere, while the values between 0.2 and 0.4 belong to a slightly stable atmosphere. The cases of each group have very similar values for the exponent. In general, it is obvious from Fig. 6 that 285 no validation cases are presented for neutral and unstable atmospheres, which would show wind shear exponents below 0.2. This is related to the fact, that measuring at unstable situations provides lower signal-to-noise ratios. Stable atmospheres are





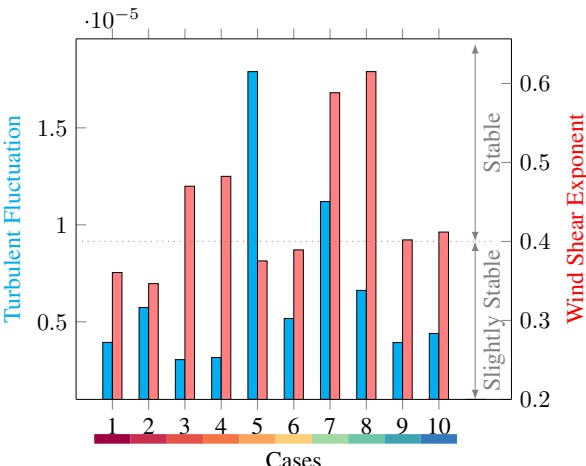

**Figure 6.** Calculated turbulent fluctuation and wind shear exponent at times of the validation cases.

predominantly developed at night, where extraneous and ambient noise is low compared to the daytime activities. Moreover, in comparison to an unstable atmosphere, the wind speed on the ground is low which reduces the wind-induced noise. This refers to wind-induced noise at the microphone and to natural wind-induced noises such as leaf rustling. Consequently, the signal-
to-noise ratio is greater with stable stratification and thus the quality of the measurement data is higher than with unstable or neutral stratification.

In addition to the exponent, the turbulent fluctuations determined according to Eq. 6 are shown in Fig. 6. Note, that no correlation is expected between the wind shear exponent and the turbulent fluctuations. Within the individual groups, these values deviate strongly from each other in some cases. In crosswind/upwind direction a turbulent fluctuation of more than
$1.75 \cdot 10^{-5}$ is given for case 5, while for similar conditions a value of about $5.2 \cdot 10^{-6}$ was calculated for case 6. Depending on the prevailing conditions, the literature values range between $10^{-5}$ and $10^{-6}$ (Salomons, 2001). Accordingly, the values calculated here are comparable to literature. In Sec. 3.3, the effect of turbulent fluctuations on sound propagation is discussed in more detail using simulation results.

The atmospheric values of temperature and relative humidity significantly determine the air absorption, whereas they have
a subordinate effect on the sound speed profiles. The averaged values as well as the standard deviation over 10 minutes are visualized for temperature and relative humidity in Fig. 7(a). As before, the pairs have similar values. With about 1 C° difference, cases 5 and 6 have the largest temperature discrepancy. Humidity differs the most between cases 1 and 2 amounting to 10%. In both cases, the high standard deviations of the humidity values of up to 45% are remarkable. A high standard deviation of the humidity might indicate frequent rainfall, so that in these cases explicit attention must be paid to the quality of
the measurement data. Lastly, the calculated values of the atmospheric air coefficient $\alpha_L$ (dB/m) for the midfrequencies of the 1/3 octave bands are visualized in Fig. 7(b) for three selected cases. As expected, the coefficient and thus the sound absorption increases with increasing frequency and decreases with increasing temperature and humidity. Case 3 is characterized by low





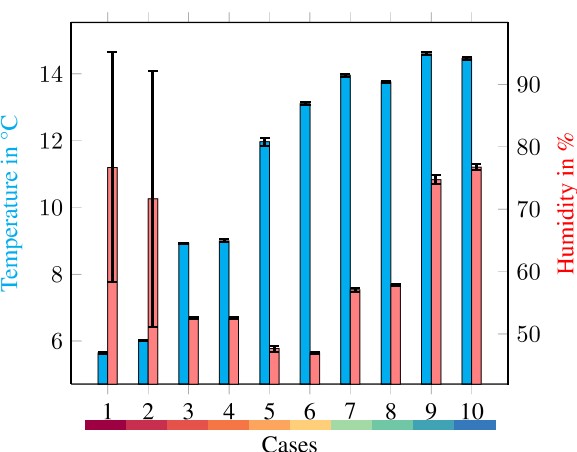

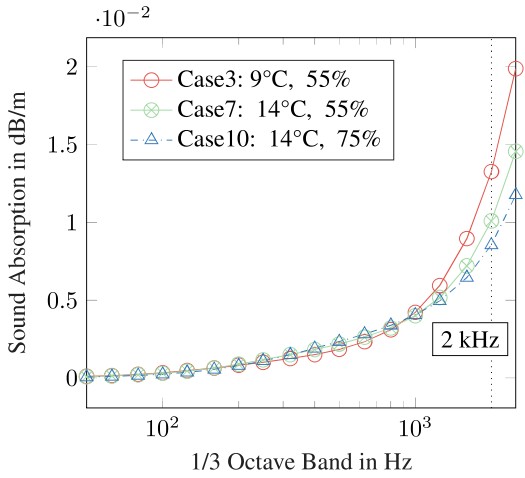

(a) Measured values and standard deviation of 10 minute data for temperature and relative humidity at 53m

(b) Calculated sound absorption for selected cases having similar values of temperature respective relative humidity

**Figure 7.** Parameters describing the atmospheric air absorption for the validation cases.

values of humidity and temperature and has a sound absorption of $0.013\ \mathrm{dB/m}$ at 2000 Hz. As a result of higher humidity and temperature, case 10 has a lower sound absorption of $0.009\ \mathrm{dB/m}$ at the same frequency.

### 3.1.2 Acoustic data

The signal-to-noise ratio (SNR) is of particular importance for the quality of the acoustic data. It represents the difference between the desired sound and background noise, i.e. in the present case the difference between the noise of the wind turbine and extraneous noise. In general, the latter includes all types of background noise, such as noise from traffic or animals, which have a significant influence on the measurements. In order to select validation cases without these significant extraneous noises, frequency-dependent selections and listening tests were performed. Frequency-dependent selection is a common methodology in which the frequency spectrum of the wind turbine is compared with spectra including extraneous noise (van den Berg, 2004; Larsson and Öhlund, 2014; Conrady et al., 2018; Martens et al., 2020).

In order to assess the background noise, recordings were conducted at the beginning of the measurement campaign during wind turbine shut down. Here, the background noise was measured with the same measurement set-up at comparable wind conditions. The measured background noise is considered to be representative for the background noise that occurred during the measurement of the validation cases. It should be mentioned that the wind turbine operations (on/off) were regulated by the power production management system. This means that the authors did not control the operational conditions of the wind turbines - neither for the turbine under investigation nor for the surrounding turbines (see Fig. 1). In times of high energy production within the whole energy system, however, the management system deliberately shuts down turbines so that recordings of background noise are possible even at operating wind speeds. Following the measurements of background noise,

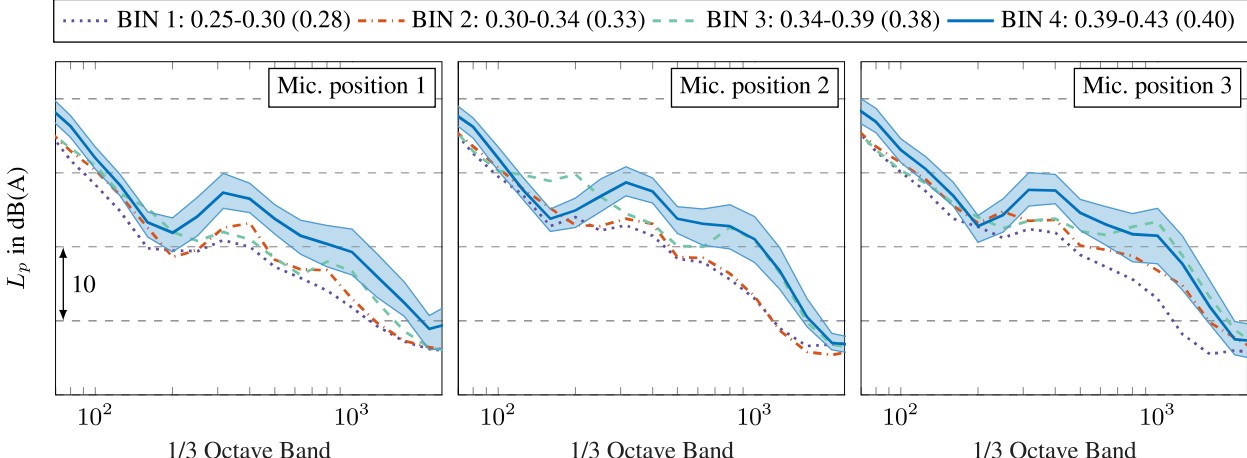

**Figure 8.** Measured background noise per 1/3 octave bands for different wind speed bins at three microphone positions. The wind speeds were recorded at a height of 100 m and are divided into bins of 1 m/s. The bins are normalized to the cut-off wind speed of the turbine. The average normalized wind speed during the background noise measurements is given in parentheses. The measured sound levels in the bins are energetic averages over the corresponding measurement period. Representative for all bins, the standard deviation of the measurements is given for BIN 4, which includes the data at the highest wind speed during the background measurements.

the authors benefited from the fact that the wind farm is also a test site. Even if surrounding turbines were switched off by the management system, the turbine under investigation continued to operate in test mode. As a result, measurements for the individual turbine were possible.

However, in Fig. 8 the measured background noise per 1/3 octave bands is shown for different wind speeds at the three microphone positions. A similar tendency is noticeable at all microphones. With increasing wind speed, the background noise increases. This trend is already known from the literature and is due to the wind-induced noises that depend on the wind speeds. High extraneous noise in the low frequency range is due to wind-induced noise at the microphone, which cannot be completely eliminated even with effective windscreens. In Fig. 8, a local peak at approx. 300 Hz is also observed. This is assumed to be wind-induced vegetation noise, such as the rustling of grasses. The peak at 1000 Hz is due to a combination of vegetation noise and the A-weighting of the sound level. In general, the standard deviation of the background measurements is higher than for the wind turbine noise. As seen in Fig. 8, the highest value is at the local level maxima with approx. 2.5 dB. Wind-induced sounds from vegetation are known to vary greatly. Consequently, it is assumed that this variation is responsible for the comparatively high standard deviations. Since the recordings were conducted before the measurement campaign, it should be noted that the background noise can change in the course of the measurement campaign depending on the vegetation. Due to prevailing wind conditions and the management system, no measurements of background noise could be taken at the end of the measurement campaign.



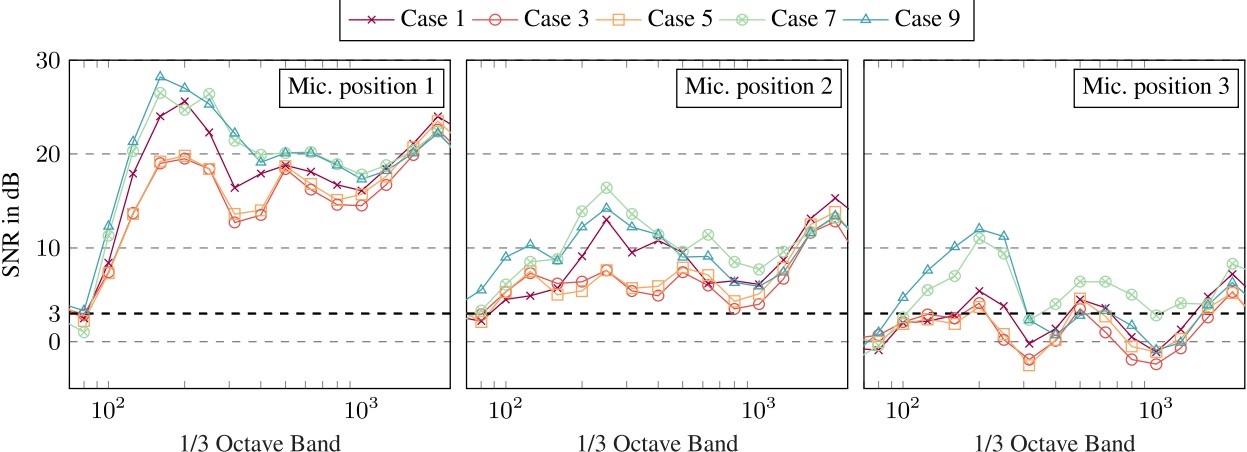

**Figure 9.** Signal-to-noise ratio per 1/3 octave bands for selected cases at three microphone positions. The background noise at BIN 4 is used for the determination.

After all, in Fig. 9 the calculated SNR is shown per 1/3 octave band at the three microphones for five selected cases, i.e. one case per group. To consider the most critical condition, the background noise at BIN 4 is used for the determination of SNR. In the guideline IEC 61400-12 (2012), three quality levels are stated. An SNR of more than 8 dB is very good and between 3 and 8 dB is sufficient. If the difference is less than 3 dB, it is recommended not to use the measurements.

In general, the SNR decreases with increasing distance, which is due to the quieter wind turbine noise. While a sufficient SNR is usually achieved at Mic. 1 and Mic. 2, the SNR at the third microphone is critical. Here, especially the values for crosswind conditions are below 3 dB and are partly in the negative range. Moreover, Mic. 1 indicates that especially very low frequencies below 100 Hz are also classified as critical. Wind-induced noise at the microphone is considered in these low frequency ranges. In addition, compared to other cases, a low SNR is calculated with the cases in crosswind condition at the first microphone position. This tendency is also observed at positions 2 and 3. The comparatively low SNR in crosswind condition is due to the source characteristics of the turbine. Because of the dipole characteristic of the trailing edge noise, a wind turbine radiates less sound in crosswind direction. Consequently, the difference to the background noise is lower in this direction.

Especially at Mic. 3 in a distance of 845 m to the wind turbine, the SNR is critical. However, to extend the database those data is also used for the validation. By carefully listening to the recordings of the validation cases, a negligible influence of the background noise on the useful signal is ensured. It should be stated that the highest wind speed bin (BIN 4) was chosen in the analysis of SNR - even though some of the validation cases were recorded at lower wind speeds. That means the worst scenario was investigated. In addition, only two measurements over a period of 5 minutes were available for BIN 4. It can be expected that the wind turbine noise is dominant at the third microphone for all cases.





## 3.2 Validation

The measurement data is used to validate the sound propagation model presented in Sec. 2.1. The measured and modeled sound propagation losses between the first and the second, respectively the third microphone position, are compared using 1/3 octave bands. For assessing the accuracy, the difference between measured and modeled propagation loss in total sound pressure
levels is examined. The modeled propagation losses include all attenuation terms introduced in Sec. 2.1, i.e. attenuation due to geometric scattering, air absorption, and other aspects such as ground effects and atmospheric refraction.

### 3.2.1 Comparison of 1/3 octave band

The comparison of measured and modeled sound propagation losses per 1/3 octave band is shown using Fig. 10 and 11. For a better overview, the standard deviations of the measured data are only given for one case of each group. In addition to the
sound propagation losses, the corresponding profiles of effective sound speed are given. Since cases 5/6 show similar results as cases 3/4 and provide no new findings, their results are given in the appendix. In the following, the differences within the groups are presented first. Subsequently, cases 1 to 8 (fig. 10) are discussed in detail. The cases with strong upwind (fig. 11) are presented at the end.

In general, the difference within the groups is very small. In groups of similar propagation conditions, very similar sound
propagation losses are measured or modeled. For example, the averaged difference in the measured values of case 1 and 2 (group 1) is between Mic 1 and Mic 2 only 0.28 dB. The averaged difference of the modeled data is 0.24 dB. Consequently, it is assumed that the accuracy within the measurements and the modeling is sufficient.

For cases 1 to 8, the measured and modeled sound propagation losses between the first and second microphone positions agree very well. In all cases, the propagation losses are at similar levels. Moreover, the peak of the losses is reproduced in the
modeling. In both the measurements and the modeling, maximum sound propagation losses are obtained at frequencies of 160 and 630 Hz. This is due to ground reflections and the subsequent interference. In all cases, the measured and modeled losses increase significantly with higher frequency at greater distances, e.g. between the first and third microphone. This is caused by the frequency-dependent air absorption.

With higher distances, i.e. for losses between 178 and 845 m, a more pronounced discrepancy between measured and
modeled values is observed - although this discrepancy is still considered to be small. Here, especially the curve characteristics between 160 and 400 Hz differ. Those differences are further analyzed in the following.

In downwind direction (cases 1 and 2), the measured peak is observed over a broader frequency spectrum when compared to the modeled spectrum. At the band with a center frequency of 400 Hz, the difference between measured and modeled values is approximately 7 dB. It is assumed that this difference is due to changed ground properties. Due to the large standard deviation
of humidity (see Sec. 3.1), an increased probability of rainfall is present during the period of Cases 1 and 2. Compared to dry grass, the propagation attributes and correspondingly the losses change with wet grass. The influence of different ground conditions on sound propagation is briefly discussed in Sec. 3.3. Note that the cases 1 and 2 were recorded at the same night and within two hours.

**Figure 10.** Comparison of measured and modeled sound propagation losses per 1/3 octave band for cases 1 to 4 and 7 to 8

left: Propagation losses between Mic 1 (178 m) and Mic 2 (535 m) including standard deviation for one measurement case

middle: Propagation losses between Mic 1 (178 m) and Mic 3 (845 m) including standard deviation for one measurement case

right: Normalized profiles of effective sound speed



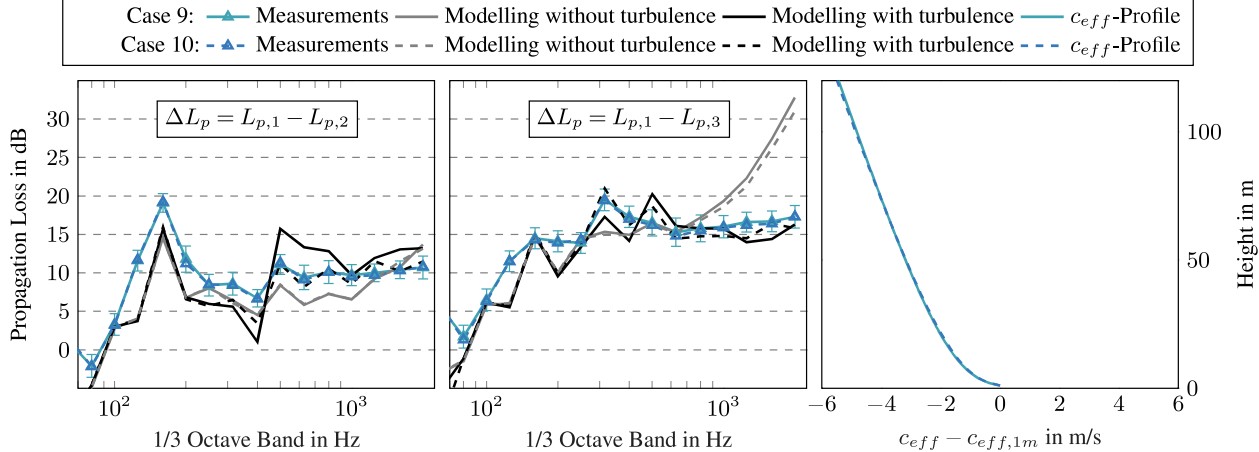

**Figure 11.** Comparison of measured and modeled propagation losses with and without turbulence per 1/3 octave band for cases 9 and 10

left: Propagation losses between Mic 1 (178 m) and Mic 2 (535 m) including standard deviation for one measurement case

middle: Propagation losses between Mic 1 (178 m) and Mic 3 (845 m) including standard deviation for one measurement case

right: Normalized profiles of effective sound speed

For crosswind conditions (cases 3 and 4), a shift of the measured peak is seen at greater distances. For the measured losses between 178 and 845 m, the peak is shifted to 250 Hz which corresponds to a shift of two bands. In the modeling, the peak is still very pronounced at 160 Hz, although a local maximum is identified at 250 Hz. However, this is much less pronounced than in the measurements. This shift might be explained by the assumptions in the PE-method, namely by the 2D-approximation (see Sec. 3.3).

For upwind conditions (cases 7 and 8), two peaks at 160 Hz and 250 respectively 315 Hz are evident in the measured and modeled losses for larger distances. With increasing distance, the refraction effects, which depend in particular on the effective sound speed profile, become more important. The sound is refracted upward in upwind conditions, while it is refracted downward in downwind conditions. Accordingly, especially at long distances, the incidence angles to the ground and thus the frequency-dependent ground reflections change in terms of the sound speed profile. As a result the curve characteristics change in dependence of sound speed profiles and hence, sound direction.

The cases 9 and 10 are characterized by very strong upwind conditions. The measured and modeled losses of these cases are shown in Fig. 11. Here, modeled results are presented without and considering turbulence. The effect of turbulence is well seen in the modeled sound propagation losses between 178 and 850 m. Due to the upward refraction, a shadow zone is created in strong upwind conditions. Since sound waves cannot enter directly in the shadow zone, the propagation losses modeled without turbulence increase significantly at high frequency ranges. The propagation losses reach up to 30 dB. In reality, however, the sound waves are scattered at turbulent eddies and consequently enter the shadow zone. Hence, these strong losses are not present in the measurements and in the modeling when turbulence is taken into account. In this work, turbulence is characterized by the Gaussian correlation function (see Sec. 2.1). As described in Salomons (2001) and Wilson et al. (1999), the Gaussian spectrum





is limited in the range of wavenumbers. This is also evident in Fig. 11. Below 200 Hz, the model results with and without turbulence are nearly identical. The turbulence characterized on the basis of the Gaussian spectrum only affects the losses in
the higher frequency range. Moreover, the turbulence in the model is generated via a random function. As a consequence, repeating modulations result in varying propagation losses. Without taking turbulence into account, the curve characteristics of cases 9 and 10 are consistent. Considering turbulence, the curves deviate from each other and provide different validation results - even if an average of fifty repetitions is used.

For case 10, the measured and with turbulence modeled losses between 178 and 535 m agree very well. In the range above
400 Hz, the modeled losses are in the range of the standard deviations of the measurements. Whereas for case 9, higher losses are predicted at frequencies above 400 Hz considering turbulence. It is assumed that with a higher number of repetitions also better agreements with measurements are achieved for case 9. In particular, since the prediction without turbulence is almost identical for case 9 and 10 and, moreover, the values of turbulent fluctuations are at a similar level being $3.9 \cdot 10^{-6}$ and $4.4 \cdot 10^{-6}$. At higher distances, clearly better validation results are achieved with turbulence than without turbulence. The
measured and modeled losses agree very well when turbulence is taken into account. This improvement is clearly evident in the high frequency range. The results with respect to turbulence as well as the sensitivity of turbulence parameters are further discussed in Sec. 3.3.

### 3.2.2  Comparison of total sound pressure level

The accuracy of the validation is assessed by comparing the total sound level losses. For this purpose, the differences between
measured and modeled total sound level losses between 178 and 535 respectively 178 and 845 m are shown in Fig. 12 and 13 for all validation cases. Accordingly, a negative value implies an overestimated and a positive value an underestimated prediction of the propagation losses.

The averaged absolute difference of the propagation losses between 178 and 535 m is less than 1 dB (0.88 dB), so that overall very good validation results are achieved. In the examined frequency range, all propagation losses are well predicted
by the model in the total sound pressure level. In Fig. 12 increasing differences with stronger upwind direction are noticed. For cases 7, 8, and 10, the difference exceeds 1 dB. It is suggested that with sufficient repeated simulations involving turbulence, good results, i.e. deviations less than 1 dB, can be obtained. With turbulence considered, the difference between measured and modeled total losses is only 0.14 dB for case 9. While in the downwind direction the model also underestimates the total sound loss, the losses in crosswind direction (Case 3-6) are overestimated at short distances.

The differences of the measured and modeled total losses between 178 and 845 m are presented in Fig. 13. In all cases, the total loss is underestimated by the model. Being 0.92 dB, the averaged absolute difference of the losses is also below 1 dB, meaning that the model generally predicts the propagation losses well also at longer distances. As already observed in the shorter distance, comparatively high differences are calculated for cases 7 and 8. In these cases already medium upwind conditions prevail. Thus, the consideration of turbulence and thereby scattering effects could be useful - especially at larger
distances. The comparatively high value in case 1 is due to the broad peak in the measurements, which is not reproduced by the model (see Fig. 10).

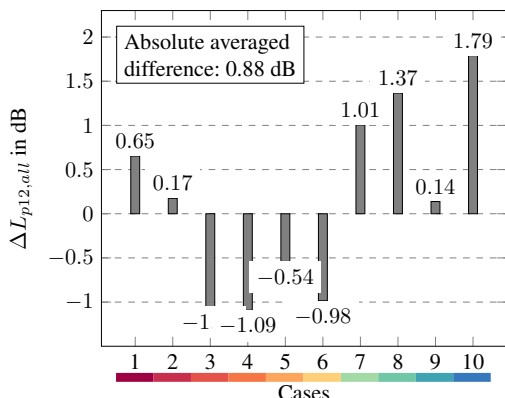

**Figure 12.** Difference of measured and modeled total propagation loss between Mic 1 (178 m) and Mic 2 (535 m)

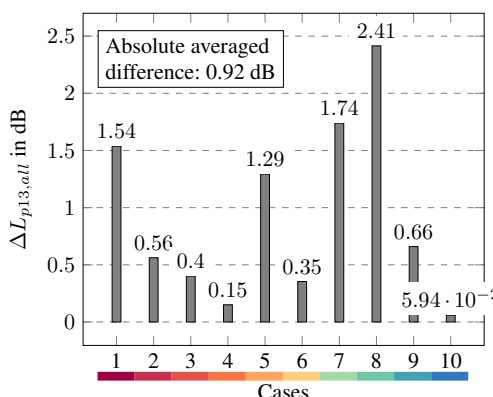

**Figure 13.** Difference of measured and modeled total propagation loss between Mic 1 (178 m) and Mic 3 (845 m)

### 3.3 Discussion

In this section, the model assumptions and limitations are first presented and discussed in context of the results. A short comparison between a monopole and a simplified wind turbine sound source is shown. Subsequently, the effects of different
input parameters are discussed in more detail. Here, different ground conditions, i.e. values of flow resistances, as well as different values of turbulent fluctuations are considered.

#### 3.3.1 Model assumptions and limitations

The PE method used has some limitations, which are described in detail in Salomons (2001). In this paper, only the most important limitations are briefly presented and discussed with regard to the application for wind turbines.





Firstly, only forward calculations are performed with the applied model, so that backscattering is neglected. Since the measurement terrain is flat and without obstacles on the propagation path, no backscattering or multiple scattering is expected in the measurements. Accordingly, this model assumption is valid for the cases to was applied to. However, for complex terrain structures, backscattering must be taken into account.

  Additionally, a 2D approach is adopted for modeling the sound propagation. Thus, azimuthal variations are neglected in

the model. This neglect is based on the axisymmetric approximation in the reduction of the wave equation from 3D to 2D. As mentioned in Sessarego and Shen (2020), in an undisturbed atmosphere and in flat terrain without obstacles, the 2D approach is valid due to the dominance of wind and temperature gradients in the vertical plane. However, in the case of a wind turbine and when considering crosswind directions, refractions in the azimuthal direction might become more important. In Cheng et al. (2009) the 2D and 3D approaches are compared for two source heights (3.4 and 68 m). According to this, at large distances

the locations of the propagation peaks and/or valleys are shifted. A shift of the propagation peaks is also observed in the validation results of this work, especially for crosswind conditions (see Fig. 10). This is suspected to be a consequence of the 2D approach.

  The implemented sound speed field is determined on the basis of meteorological measurements in the center of the wind farm. Hereby, vertical wind components are neglected, which is another limitation of the methodology used. The calculation

of the effective sound speed is based on the measured horizontal wind speeds. According to the literature, the consideration of vertical wind components is recommended when examining high sources such as wind turbines. In addition, the implemented sound speed field is constant over the distance and is only variable in the height so that further limitations can be concluded. For example, the wake of the wind turbine is not estimated from the measurements and no wake is considered in the simulations. According to Heimann et al. (2011) and Barlas et al. (2017b), considering the wake leads to higher levels in the far field.

Herein, the level depends on the turbulence intensity and the wake length. Especially during stable atmospheric conditions, the neglect of wake effects leads to an underestimation of modeled sound pressure level (Barlas et al., 2017b).

  Lastly, a simplified wind turbine sound source is used for the modelling. A comparison between this and a monopole sound source at hub height is shown in Fig. 14 for the first validation case. Compared to the results of the measurements and the modelling with a wind turbine sound source, multiple peaks and valleys are observed in the 1/3 octave spectrum

of the simulation results with a monopole sound source. These are caused by interference effects and are smoothed out in reality by superimposing the interference patterns of several sources. Compared to the wind turbine source, the model results of the monopole sound source reproduce the 1/3 octave spectrum of propagation losses less well and deviate more from the measurements. In terms of accuracy and computational cost, the applied approach is a good and efficient method for sound source modeling of a wind turbine. As result the assumption seems reasonable.

### 3.3.2 Effect of input parameters

The influence of selected input parameters on the modeled propagation losses is briefly discussed. With regards to the validation results, the influence of different ground properties as well as different turbulence intensities in the atmosphere is studied.





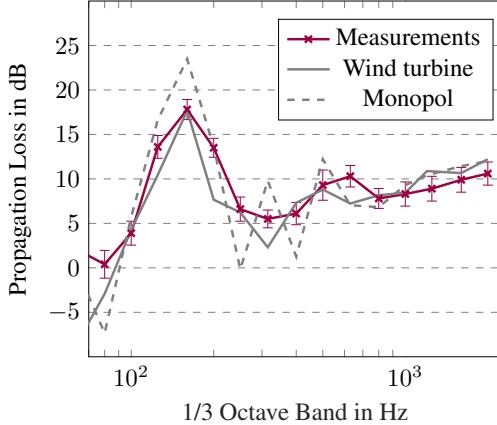

**Figure 14.** Comparison of measured and modeled sound propagation losses with a wind turbine sound source and a monopole source per 1/3 octave band between 178 and 535 m for case 1

**Table 4.** Classification of selected ground impedance types i.e. values of flow resistivity according to Plovsing and Kragh (2000)

| Class | Representative flow resistivity $\sigma$ in $kNsm^{-4}$ | Description |
|-------|--------------------------------------------------------|-------------|
| A | 12.5 | Very soft (snow or moss-like) |
| C | 80 | Uncompacted, loose ground |
| D | 200 | Normal uncompacted ground |
| E | 500 | Compacted field and gravel |

In the Delany-Bazley model, the ground impedances are only related to the flow resistivity. Hence, the influence of ground properties on the sound propagation is examined by selected representative values of flow resistivity which are based on the classification given in Plovsing and Kragh (2000). The classification including the values of flow resistivity and a description are summarized in Tab. 4. Accordingly, values from 12.5 $kNsm^{-4}$ (very soft) to 500 $kNsm^{-4}$ (compacted field) are considered and discussed. This discussion is representatively performed on the example of the first validation case and the propagation losses between 178 and 535 m. For this purpose, measured and simulated losses at different ground impedances are plotted per 1/3 octave band in Fig. 15.

The selected ground impedances have an impact on the level of propagation losses as well as on the position and width of the propagation peak. Generally, lower flow resistivity results in higher propagation losses. This is due to the increased absorption and/or decreased reflections at the ground for lower values of flow resistivity. Consequently, higher propagation losses are modeled with moss-covered ground (12.5 $kNsm^{-4}$) than with compacted fields (500 $kNsm^{-4}$). In addition, a broader propagation peak with decreasing values is observed in Fig. 15. This phenomenon is not caused by the increasing





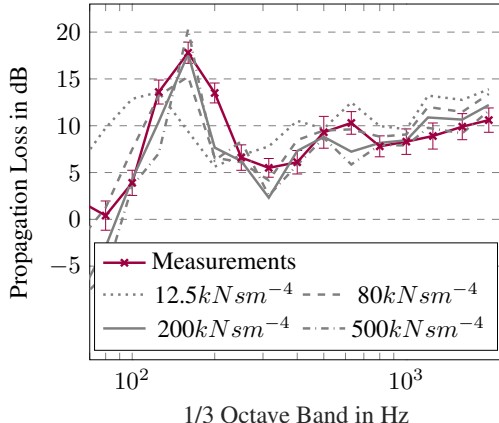

**Figure 15.** Comparison of measured and modeled sound propagation losses with representative values of flow resistivity per 1/3 octave band between 178 and 535 m for case 1

absorption but by a phase shift between the incident and the reflected wave. At very low values like 12.5 $kNsm^{-4}$ the peak is shifted towards lower frequencies. With a compact field, the propagation peak is clearly present at 160 Hz. The nearby losses at 125 and 200 Hz are lower by almost 15 dB. In comparison, the propagation peak is broader for uncompacted, loose ground. The difference between the modeled losses at 125 and 160 Hz is less than 5 dB. This is particularly interesting with respect to case 1/2 and case 7/8. Compared to other validation cases, broader propagation peaks are measured here, and are not reproduced

by the model having a ground conditions of 200 $kNsm^{-4}$. Since ground conditions are variable over the measurement time and no ground measurements were conducted, the ground properties cannot be determined precisely. It is assumed that with a precise determination of the ground properties which can change along the propagation path, the broad measured propagation peak can be better modeled. Precise ground information on the propagation path would result in better validation results.

In order to examine the impact of turbulent fluctuations on sound propagation in the upwind direction, additional simulations

are carried out for validation case 10. Here, turbulent fluctuations of $\mu = 10^{-5}, 5 \cdot 10^{-6}$ and $10^{-6}$ are considered and compared with the measured and the previously modeled results with $\mu = 4.4 \cdot 10^{-6}$. The corresponding sound propagation losses per 1/3 octave band are shown in Fig. 16. Generally, the frequencies at which the propagation peaks and valleys are apparent do not change with varying turbulence values. In the frequency range below 200 Hz, varying values result in minor changes of sound levels. That is due to the fact that the Gaussian correlation spectrum only effects the high frequency range. The impact

of turbulence increase with increasing frequency. As illustrated in the figure, low turbulence values such as $\mu = 10^{-6}$ lead to higher propagation losses in the high frequency range. In the case of the shown example, the best validation results are given by the value for turbulent fluctuations determined on the basis of the measurements, i.e. $\mu = 4.4 \cdot 10^{-6}$ . From these investigations, it can be assumed that the available meteorological data, i.e. the wind speed and temperature fluctuations, are sufficient for the determination of atmospheric turbulence. Note that still better results could be obtained by increasing the

number of realizations. In this work, 50 realizations were carried out.



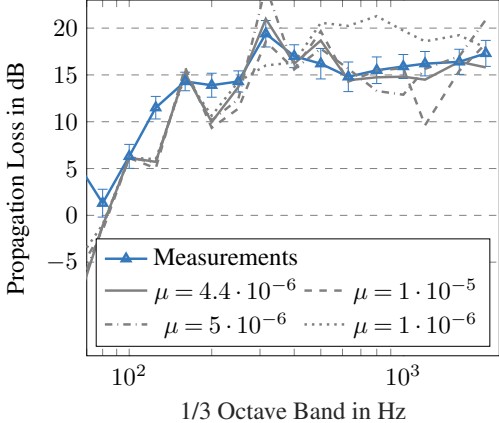

**Figure 16.** Comparison of measured and modeled sound propagation losses with representative values of turbulent fluctuations per 1/3 octave band between 178 and 845 m for case 10

## 4 Conclusion and outlook

The objectives of this paper were to introduce, to prepare, to apply and to provide comprehensive wind turbine noise measurement data for the systematic validation of sound propagation models. Extensive measurement campaigns were carried out in the area of wind turbines, which involved the acquisition of meteorological, acoustic and turbine-specific data. Meteorological

quantities, such as wind speed, temperature and humidity, were collected at different heights of a 100m measurement mast. For the recording of acoustic data, autarkic acoustic measuring stations were positioned at 178, 535 and 845 m from the wind turbine.

The atmospheric and the acoustic quantities were processed and analyzed. Based on this, a total of 10 validation cases were identified, which were divided into five groups depending on the direction of propagation. Based on the meteorological

measurements as well as the SCADA data of the turbine, relevant input parameters for sound propagation models were derived. In addition to the measurement geometry and information on the determination of air absorption, these also include sound speed profiles and turbulence parameters.

In this paper, the processed measurement data is used to validate a sound propagation model that applied the Crank-Nicolsen parabolic equations method. The validation was performed by comparing the measured and modeled sound propagation losses

per 1/3 octave spectra and total sound pressure levels between 178 and 535 and 845 m, respectively.

In all wind directions, the measured spectrum is well reproduced by the model. In both measurements and modeling, losses increase with increasing frequency due to air absorption. Because of interferences, peaks and valleys of the propagation losses exist in the frequency band and are identified in measured and modeled data. In upwind direction very good results are obtained when considering turbulence parameters. A comparison between the measured and modeled total sound level losses show a





good agreement. In considered distances, the average deviation between the losses is less than 1 dB. At greater distances the total sound level loss is slightly underestimated by the model in all wind directions.

A brief parameter study on the effects of input parameters shows that better results can be obtained with a even more accurate reproduction of the measurement conditions. Discrepancies in the 1/3 octave spectrum might be eliminated by a distance-dependent implementation of ground parameters. Also the consideration of wake effects is reasonable.

This paper provides the first step towards the publication of measurement and simulation data in the field of wind turbine sound propagation. The data sets used for the validation are provided for further research purposes. In further steps more data sets will be added in the future. A comprehensive structured data repository will be created, containing anonymized research data concerning the sound emission and immission from wind turbines under various atmospheric and operational conditions.





*Data availability.* The measured and modeled data used for the validation are provided and available for research purposes. URL: https:
//data.uni-hannover.de/dataset/wtn-propagation

## Appendix A:  Validation results - Comparison of 1/3 octave band for cases 5 and 6

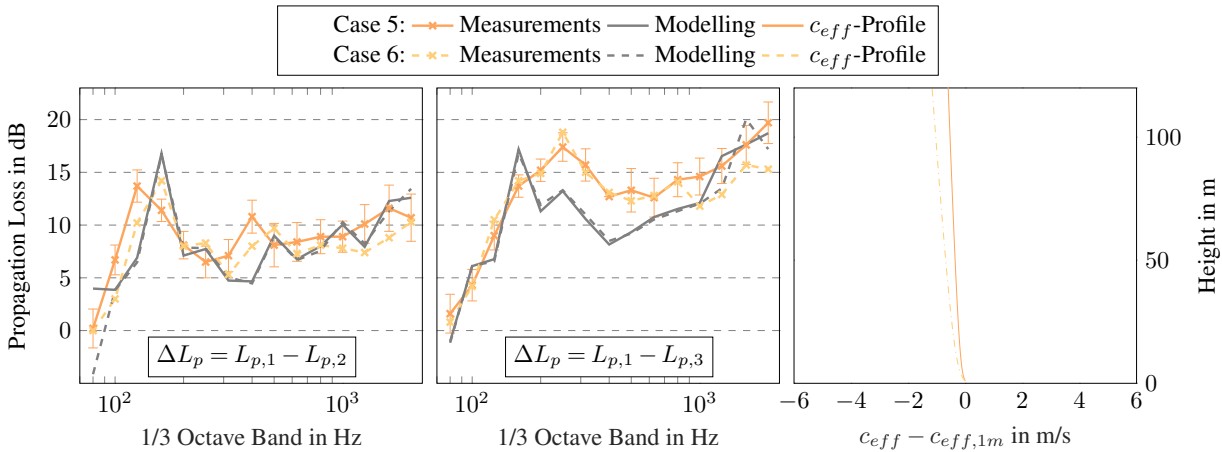

**Figure A1.** Comparison of measured and modeled propagation losses per 1/3 octave band for cases 5 and 6

left: Propagation losses between Mic 1 (178 m) and Mic 2 (535 m) including standard deviation for measurement case 5

middle: Propagation losses between Mic 1 (178 m) and Mic 3 (845 m) including standard deviation for measurement case 5

right: Normalized profiles of effective sound speed

*Author contributions.* SK did the main research work, performed and analyzed the field measurements, conducted the validation and wrote most of the manuscript. The numerical model was implemented by JH. Through discussions and feedback, JH, TB and RR contributed to the interpretation and discussion of the results. The manuscript was revised and improved by all authors.

*Competing interests.* RR is member of the editorial board of Wind Energy Science. The peer-review process was guided by an independent editor. The authors have also no other competing interests to declare.

*Acknowledgements.* Within in the project "WEA-Akzeptanz", the research at Leibniz University of Hannover is funded by the Federal Ministry for Economic Affairs and Energy by an act of the German Parliament (project ref. no. 0324134A). The Institute of Structural Analysis is part of the Center for Wind Energy Research For-Wind. The authors gratefully acknowledge the financial support from the



research funding organization, the provision of meteorological data by the DNV GL (Det Norske Veritas and Germanischer Lloyd), and the great support from the operator of the wind farm, named by Bürgerwindpark Janneby eG. For further information about the project, please visit the project homepage at www.wea-akzeptanz.uni-hannover.de.





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
