# Peer review of "A new base of wind turbine noise measurement data and its application for a systematic validation of sound propagation models"

_Wind Energy Science, 2021_

## Referee Comment (RC2)

**Review of the article "A new base of wind turbine noise measurement data and its application for a systematic validation of sound propagation models"**

March 23, 2022

**1 Specific comments**

**1.1 Source modeling**

You mention page 4 that you follow the approach of Barlas et al. (2017b) to represent the wind turbine as an acoustic source. You indeed consider 3 source heights located at $h_s = h - 0.85l$, $h_s = 0$ and $h_s = h + 0.85l$, with $h$ the hub height and $l$ the rotor length, as in the steady case considered by Barlas et al. (2017b). You then write line 125 that "simulation results are logarithmically summed (Barlas et al., 2017b)". When I read the article of Barlas et al. (2017b) I don't understand that they summed the simulation results for the 3 source heights. Indeed Figure 9 of this paper shows the influence of the source height on this predicted sound pressure levels, but no mention is made of a logarithmic sum except over frequency bands in Equation (2). For me this way of calculating the sound pressure level $L_p$ and then the propagation loss $\Delta L_p$ has not been done before, and deserves a to be discussed in details.

Following Equation (1), the sound pressure level is calculated as:

$$L_p(f) = L_W(f) - 10\log(4\pi R^2) - \alpha_L(f)R + \Delta L(f). \tag{1}$$

Although not clearly stated in the article, this model is valid for one point source (monopole). If the wind turbine is modeled with one point source, it is straightforward to calculate the propagation loss following Equation (12):

$$\Delta L_p = L_{p,1} - L_{p,m} = -10\log\left(\frac{R_1}{R_m}\right) - \alpha_L(R_1 - R_m) + \Delta L_1 - \Delta L_m. \tag{2}$$

Note that it is not needed to assume that the sound power level $L_W$ is equal to zero as you do on page 10. The term $L_W$ is canceled when calculating the propagation loss.

Now if you model the wind turbine with 3 source heights (or more), Equation (1) becomes:

$$L_p^n(f) = L_W^n(f) - 10\log(4\pi(R^n)^2) - \alpha_L(f)R^n + \Delta L^n(f), \quad n = 1..3, \tag{3}$$

where the subscript $n$ refers to the point source number. Then if you sum logarithmically the contributions from the 3 point sources, we obtain:

$$L_p(f) = 10\log_{10}\left(10^{L_p^1/10} + 10^{L_p^2/10} + 10^{L_p^3/10}\right). \tag{4}$$

The propagation loss is now given by:

$$\Delta L_p = 10\log_{10}\left(10^{L_{p,1}^1/10} + 10^{L_{p,1}^2/10} + 10^{L_{p,1}^3/10}\right) - 10\log_{10}\left(10^{L_{p,m}^1/10} + 10^{L_{p,m}^2/10} + 10^{L_{p,m}^3/10}\right)$$

$$= 10\log_{10}\left(\frac{10^{L_{p,1}^1/10} + 10^{L_{p,1}^2/10} + 10^{L_{p,1}^3/10}}{10^{L_{p,m}^1/10} + 10^{L_{p,m}^2/10} + 10^{L_{p,m}^3/10}}\right)$$

If the values of the sound power level $L_W^n$ for the 3 point sources is unknown, it does not seem possible to simplify this expression. Your solution is to set $L_W^n = 0$, but this is not physically valid. I think your approximation consists in reality in distributing equally the sound power level among the 3 point sources: $L_W^n = L_W/3$. In this case the sound power level terms cancel when the propagation loss is calculated.

Now is this approximation valid? It is difficult to answer this question without comparing the simplified model to a reference solution involving a real blade in rotation. I nevertheless feel that giving an equal power to each point source may yield too much strength to the top and bottom point sources, as one blade spends twice as much time close to the hub than at each extremity. Furthermore, other effects may change the source distribution over height such as the effects of directivity and convective amplification [1], or the effect of inhomogeneous inflow due for instance to wind shear [2].

**1.2   Description of measurement setup and data processing**

The description of the experimental setup in Section 2.2 raises several issues:

- Figure 1 shows that the wind farm where the noise is measured is composed of 6 wind turbines. You are interested in the noise of a specific wind turbine located on the southern part of the farm, but Microphone 2 is close to another wind turbine, and microphone 3 seems relatively close to yet another wind turbine. Does it mean that you have selected periods where only one wind turbine in the park is on?

- In Figure 2, you write that the hub height is $93\,\mathrm{m}$ and the rotor diameter is $114\,\mathrm{m}$. However in line 254, you write that the hub height is $119\,\mathrm{m}$ and the rotor diameter is $114\,\mathrm{m}$. Finally in the readme.pdf that is provided along with the data on the project homepage, the hub height is $114\,\mathrm{m}$ and the rotor diameter is $93\,\mathrm{m}$! Please correct these differences...

- In order to measure temperature, wind speed and wind direction at different heights, you use the data of a meteorological mast located a few kilometers away from the wind turbine.

    - Since you have access to SCADA recordings, have you compared the wind speed and direction on the wind turbine hub with the ones measured by the anemometer on the meteo mast. It would allow you to check the spatial homogeneity of atmospheric quantities in this specific wind farm.

    - What is the resolution of the temperature sensors you use? If this is $1\,\mathrm{Hz}$ as for the wind speed (line 170) it is likely to be insufficient to characterize turbulent fluctuations (see subsection below on atmospheric turbulence).

- Page 15, you mention that background noise can change in the course of the measurement campaign depending on the vegetation. What is the duration of the measurement campaign? Could you add the date and hour of each case in Table 3? This would also be a valuable information regarding the change in ground impedance (see subsection below).

Furthermore, there are some lack of details on the data processing that you perform:

- You calculate the effective sound speed in Equation (8) using $c_{\mathrm{eff}} = c_0 - |u|\cos\gamma$. If $\gamma$ is the angle between the wind direction and the sound propagation direction, with $\gamma = 0$ corresponding to downwind conditions, the expression should be $c_{\mathrm{eff}} = c_0 + |u|\cos\gamma$.

- You explain line 234 that a sampling rate of 5 Hz is used within the 1/3 octave bands. Is it true only for the measured data? Do you calculate the numerical SPL only at the center frequencies of the 1/3 octave band or you consider several frequencies per 1/3 octave band?

- In Figure 4(a) the values of wind speed cannot be read. Also, why don't you plot values at heights closer to the hub (such as 100 m for wind speed and 95 m for wind direction)? They would be more representative of what the average inflow impinging on the wind turbine.

- Figure 5 does not provide any additional information compared to Figure 4(b) and can be removed. The values of RMSE can simply be given in the text.

**1.3 Characterization of atmospheric turbulence**

To characterize the turbulent fields of temperature and wind speed, you use a Gaussian correlation function with a correlation length of 1 m and a variance of index of refraction fluctuations $\mu^2$. This variance is related to the variance of the temperature and wind speed turbulent fluctuations, as given by Equation (6) of your paper.

To study the scattering of acoustic waves by turbulent structures, it is however necessary to measure temperature and wind speed fluctuations with a sampling frequency of at least 20 Hz (with an ultrasonic anemometer for instance). Indeed, the structures of interest are generally located in the inertial subrange (see Figures 1 and 3 of Wilson et al. [3]). If you use a sampling frequency of 1 Hz, you will only capture turbulent structures in the energy subrange, as can be seen for instance in Figure 8 of Daigle et al. [4].

Figure 8 of Daigle et al. [4] also clearly shows that the Gaussian model is not adapted to capture the spectral slope in the inertial subrange. At best, it can fit the spectrum over a small range of turbulence wavenumbers, as explained in Wilson et al. [3], but the von Kármán model is generally recommended.

Thus, I disagree with your sentence lines 518-519, where you conclude that "the available meteorological data, i.e. the wind speed and temperature fluctuations, are sufficient for the determination of atmospheric turbulence". I think that the experimental estimates of $\mu^2$ cannot be used to calculate the turbulence spectrum. The good agreement that you obtain in Figure 11 at 845 m with turbulence above 1000 Hz may not be meaningful. Indeed the SPL values are very low in the shadow zone, and corresponding signal-to-noise ratio is very low, as can be seen in Figure 9. Thus the difference between the model predictions without turbulence and the measurements above 1000 Hz is likely not be due (at least partly) to background noise.

**1.4 Ground effect**

You show clearly in Figure 15 that the flow resistivity used to calculate the ground impedance in the Delany-Bazley-Miki model has a significant impact on the propagation loss. From Figure 10 and Figure A.1, it appears that the modeled and measured third octave band spectra are in good agreement at 535 m, but not at 845 m, with differences greater than 5 dB between 250 Hz and 500 Hz approximately. The main reason for the discrepancies between model predictions and measurements is likely to be the ground effect, as refraction effects cannot explain the differences (small vertical sound speed gradient for cases 3/4 and cases 5/6).

You mention that ground properties may have changed between the various cases lines 389-394, due for instance to the soil humidity. It has indeed been shown in the literature that the flow resistivity can vary drastically if the soil is dry or wet. In Figure 15 you have

tested various flow resistivities at 535 m. Could you perform the same tests at 845 m where the agreement is less good, to test if a change in flow resisitivity could explain the differences between model predictions and measurements?

Also, you show in Figure 14 that model predictions with a single point source yield strong interference minima and maxima, that are smoothed out when you consider your wind turbine model with 3 point sources, as shown by various researchers [5,6]. However at 845 m the interference minima and maxima associated with ground effect seem too strong in the model predictions compared to the measurements. This could mean that 3 point sources is not enough to calculate the propagation loss. Could you test with a greater number of point sources distributed over the rotor height?

**1.5 Section 3.2.2 on total SPL**

In section 3.2.2, you discuss in length the agreement between measured and modeled propagation losses averaged over all frequencies. It seems to me that this is a bad indicator to check the validity of the model. For instance for case 4 at 845 m, you obtain a small difference of 0.15 dB, while the spectra in Figure 10 differ significantly. What happens is that the overestimation at low frequencies is compensated by the underestimation at high frequencies. This is even more problematic when you write lines 441-442: "Being 0.92 dB, the averaged absolute difference of the losses is also below 1 dB, meaning that the model generally predicts the propagation losses well also at longer distances." This averaging makes no sense to me! If you want to quantify the validity of your model predictions using a single number, my suggestion would be to calculate the mean over frequencies of the absolute value of the differences (or something similar):

$$\text{Mean Difference} = \frac{1}{N} \sum_i^N |\Delta L_{p,i}(\text{model}) - \Delta L_{p,i}(\text{measurement})|. \tag{5}$$

**1.6 Section 3.3.1 on model assumptions and limitations**

In the discussion section 3.3.1, you insist on two assumptions that have a minor influence on the model predictions in my opinion:

1. 2D approach: based on the article of Cheng et al. (2019), you explain the differences between model predictions and measurements crosswind by the fact that azimuthal refraction is neglected. In the study of Cheng et al., some differences can indeed be seen in their figures 10 and 11 in the cross-wind direction between 2D and 3D aproaches (5 dB in the worst case for a source close to the ground). However this is for an unrealistic wind speed profile; the wind speed is greater than 40 m/s above a height of 70 m! For more realistic wind speed profiles, Salomons (2001) has shown in Figures 4.14 and 4.15 that the 2D approximation is valid with a good accuracy.

2. Vertical components of wind speed: you write lines 470-471 that "According to the literature, the consideration of vertical wind components is recommended when examining high sources such as wind turbines." It would be good to cite relevant articles to show that this effect is important, as neglecting the vertical components of wind speed is generally considered as a good approximation for outdoor noise propagation.

3. Wake effect: it has been shown that sound propagation through a wind turbine wake can have a significant effect on the SPL, however this is limited to downwind configurations. As you don't consider angles between $-30°$ and $30°$, wake effect is unlikely to be important in your study.

In my opinion, the discussion section should focus on the main issues that are:

1. the source model (see above subsection);

2. the ground characteristics that are not measured and can only be found by comparing model predictions and measurements (see above subsection);

3. the characterization of atmospheric turbulence (see above subsection);

4. the possible changes of temperature and wind speed profiles with distance (the meteo mast being relatively far from the wind turbine under study).

The conclusion should be changed accordingly, for instance in lines 543-544: "Discrepancies in the 1/3 octave spectrum might be eliminated by a distance-dependent implementation of ground parameters. Also the consideration of wake effects is reasonable."

**2 Technical corrections**

Here is a list of suggested corrections:

1. abstract line 10: "1/3 octave spectra" $\Rightarrow$ "third octave band spectra"

2. Introduction line 20: the article of Bérengier et al. is about outdoor noise propagation models in general, there is no application to wind turbine noise.

3. Introduction line 32: the conference paper by Shen et al. was published in 2020, not in 2005.

4. Introduction lines 38-40: when the difference between measurements and model predictions are between -3.4 and 2.5 dB, I don't think it can be concluded that the general agreement is good. Furthermore, from what I understand from Shen et al. (2020) the comparison is not done exactly crosswind, as they write in Section 2.3 that the wind directions are 269 to 279 degrees while the microphone line direction is 225 degrees (44 to 54 degrees respective to downwind).

5. Introduction lines 46-47: "Moreover, with a hub height of 60 m, the investigated wind turbine does not correspond to the current scales of up to 165 m". It is uncommon to encounter onshore wind turbines with a hub height of 165 m, It is typically between 80 m and 120 m.

6. Introduction lines 51: "measurement mast" $\Rightarrow$ "meteorological mast"

7. Introduction lines 53-59: when you discuss the results of Sonbdergaard and Plovsing (2009), I think it is not sufficient to provide the average deviation between model predictions and measurements. For the validation cases with a single wind turbine, the differences are between -3.8 dB and 1.3 dB, which corresponds to an average deviation of -1.0 dB and a standard deviation of 2.3 dB. I suggest you add the standard deviation or that you give the extreme values.

8. Equation (1) page 3: the term associated to atmospheric absorption is usually written $-\alpha_L R$ and not $-\alpha_L R/1000$. See Equation (3.8) of Salomons (2001).

9. Line 116 page 4: I think the coefficients $A_0$, $A_2$ and $B$ in Equation (3) have been obtained by an optimization procedure based on an analytical solution. Thus they are not obtained from empirical studies.

10. line 169 page 6: you write "having a resolution of ten minutes". I think you mean that the data is averaged over ten minutes, but the data resolution is lower.

11. line 179 page 7: I would replace "characterizing the turbulences" by "characterizing the turbulent fluctuations".

12. Lines 308-309 and Figure 7(b): it would be easier to read values of the absorption coefficient in dB/100m or dB/km.

13. Line 319 page 15: I think "However" should be removed.

**References**

[1] S. Oerlemans, P. Sijtsma and B. Mndez Lpez, Effect of wind shear on amplitude modulation of wind turbine noise , *Journal of Sound and Vibration*, vol. 299, pp. 869–883, 2007.

[2] S. Oerlemans, Effect of wind shear on amplitude modulation of wind turbine noise , *International Journal of Aeroacoustics*, vol. 14, pp. 715–728, 2015.

[3] D.K. Wilson, J.G. Brasseur, and K.E. Gilbert, Acoustic scattering and the spectrum of atmospheric turbulence, *Journal of the Acoustical Society of America*, vol. 105, pp. 30–34, 1999.

[4] G. A. Daigle, T. F. W. Embleton, and J. E. Piercy, Propagation of sound in the presence of gradients and turbulence near the ground, *Journal of the Acoustical Society of America*, vol. 79, pp. 613–627, 1986.

[5] K. Heutschi, R. Pieren, M. Müller, M. Manyoky, U.W. Hayek, K. Eggenschwiler, Auralization of Wind Turbine Noise: Propagation Filtering and Vegetation Noise Synthesis, *Acta Acustica united with Acustica*, vol. 100, pp. 13–24, 2014.

[6] B. Cotté, Extended source models for wind turbine noise propagation, *Journal of the Acoustical Society of America*, vol. 145, pp. 1363–1371, 2019.

---

## Author Comment (AC1)

**Response to reviewer's comments**

We apologize for the late response. Due to parental leave and health issues, earlier editing was not possible.

We would like to thank the reviewers for this detailed review and valuable comments. They have greatly improved the quality of the paper. We agree that data publication is very important. We are happy that the data is already used in the *IEA Wind Task 39 WP3* to validate further models. Because of this, misunderstandings in the data and its publication could have been corrected. As a result, the modelling results changed which is only noticeable in strong upwind direction. The general outcome of the paper did not change.

This document provides the answers to the reviewers' comments. The lines and equations used in the answers refer to the new version. The modified paragraphs are marked in the manuscript. Purple markers in the manuscript indicate author changes that were made independently of the review. Only subject-related changes are marked here. Linguistic improvements, for example, are not highlighted.

**Reviewer 1:**

Thank you for the positive assessment and the recommendation.

**Reviewer 2: (marked blue in the manuscript)**

Thank you very much for the detailed review of our paper. We appreciate the effort and are grateful for the valuable comments.

We have not been able to identify the cause of the discrepancy between the modelled and measured 1/3 octave spectra. We suspect that the discrepancy is due to source and ground modelling. Unfortunately, we have no information about the precise source and its radiation. We are also limited by the fact that no ground measurements were made, and that the ground model uses a constant value for the entire propagation path. We hope that researchers who apply a more detailed model will use the data and provide additional information.

We have discussed the modelling of atmospheric turbulence extensively and have decided to exclude atmospheric turbulence from the paper. Consequently, turbulence is not included in strong upwind simulations.

The review is commented in more detail below.

**1. Specific comments**

**1.1 Source modelling**

You mention page 4 that you follow the approach of Barlas et al. (2017b) to represent the wind turbine as an acoustic source. You indeed consider 3 source heights located at  $h_s = h - 0.85l$ ,  $h_s = 0$  and  $h_s = h + 0.85l$ , with h the hub height and l the rotor length, as in the steady case considered by Barlas et al. (2017b). You then write line 125 that "simulation results are logarithmically summed (Barlas et al., 2017b)". When I read the article of Barlas et al. (2017b) I don't understand that they summed the simulation results for the 3 source heights. Indeed Figure 9 of this paper shows the influence of the source height on this predicted sound pressure levels, but no mention is made of a logarithmic sum except over frequency bands in Equation (2). For me this way of calculating the sound pressure level  $L_p$  and then the propagation loss  $L_p$  has not been done before, and deserves a to be discussed in details. Following Equation (1), the sound pressure level is calculated as:

$$L_p(f) = L_W(f) - 10 \cdot \log 4\pi(R)^2 - \alpha_L(f) \cdot R + \Delta L(f).$$
(0.1)

Although not clearly stated in the article, this model is valid for one point source (monopole). If the wind turbine is modeled with one point source, it is straightforward to calculate the propagation loss following Equation (12):

$$\Delta L_p = L_{p,1} - L_{p,m} = -10\log\left(\frac{R_1}{R_m}\right) - \alpha_L(f)(R_1 - R_m) + \Delta L_1 - \Delta L_m \tag{0.2}$$

Note that it is not needed to assume that the sound power level LW is equal to zero as you do on page 10. The term  $L_W$  is canceled when calculating the propagation loss. Now if you model the wind turbine with 3 source heights (or more), Equation (1) becomes:

$$L_p^n(f) = L_W^n(f) - 10 \cdot \log 4\pi (R^n)^2 - \alpha_L(f) \cdot R^n + \Delta L(f)^n, n = 1...3,$$
(0.3)

where the subscript *n* refers to the point source number. Then if you sum logarithmically the contributions from the 3 point sources, we obtain:

$$L_p(f) = 10\log_{10}\left(10^{L_p^1/10} + 10^{L_p^2/10} + 10^{L_p^3/10}\right).$$
(0.4)

The propagation loss is now given by:

$$\Delta L_{p} = 10 \log_{10} \left( 10^{L_{p,1}^{1}/10} + 10^{L_{p,1}^{2}/10} + 10^{L_{p,1}^{3}/10} \right) - 10 \log_{10} \left( 10^{L_{p,m}^{1}/10} + 10^{L_{p,m}^{2}/10} + 10^{L_{p,m}^{3}/10} \right)$$
$$= 10 \log_{10} \left( \frac{10^{L_{p,1}^{1}/10} + 10^{L_{p,1}^{2}/10} + 10^{L_{p,m}^{3}/10}}{10^{L_{p,m}^{1}/10} + 10^{L_{p,m}^{2}/10} + 10^{L_{p,m}^{3}/10}} \right)$$

If the values of the sound power level  $L_W^n$  for the 3 point sources is unknown, it does not seem possible to simplify this expression. Your solution is to set  $L_W^n = 0$ , but this is not physically valid. I think your approximation consists in reality in distributing equally the sound power level among the 3 point sources:  $L_W^n = L_W/3$ . In this case the sound power level terms cancel when the propagation loss is calculated.

Now is this approximation valid? It is difficult to answer this question without comparing the simplified model to a reference solution involving a real blade in rotation. I nevertheless feel that giving an equal power to each point source may yield too much strength to the top and bottom point sources, as one blade spends twice as much time close to the hub than at each extremity. Furthermore, other effects may change the source distribution over height such as the effects of directivity and convective amplification [1], or the effect of inhomogeneous inflow due for instance to wind shear [2].

Thank you for the very detailed review on the sound source description. The source modelling follows the work of Nyborg et al. (2022). Three incoherent sources with equal source strength are assumed. We totally agree that it is a strong simplification. In reality, it is unlikely that all sources have the same strength. However, the simplification is necessary because of the lack of information about the individual point sources and the distribution of the sources over height. A detailed analysis of the radiation conditions is beyond the scope of this paper. The focus of the paper is on the measurements. However, the issue has been addressed in the paper (l. 94-104; l. 133-141).

**1.2 Description of measurement setup and data processing**

*Figure 1 shows that the wind farm where the noise is measured is composed of 6 wind turbines. You are interested in the noise of a specific wind turbine located on the southern part of the farm, but Microphone 2 is close to another wind turbine, and microphone 3 seems relatively close to yet another wind turbine. Does it mean that you have selected periods where only one wind turbine in the park is on?*

Thank you, this is an important point that needs to be clarified in the paper. During the selected periods, only one turbine in the park is on. In northern Germany, there are situations, especially at night and with high wind speeds, where more energy is produced than is needed. In these situations, turbines are turned off. Since the wind farm is a test site, it was possible to register a test for the turbine under investigation. As a result, this turbine continued to operate in such situations while the others were shut down. During the data evaluation, the shutdown of the other turbines was controlled by SCADA data. In the paper it is now clear that only one turbine is on during the selected periods (l. 150-152).

In Figure 2, you write that the hub height is 93 m and the rotor diameter is 114 m. However in line 254, you write that the hub height is 119 m and the rotor diameter is 114 m. Finally in the readme.pdf that is provided along with the data on the project homepage, the hub height is 114 m and the rotor diameter is 93 m! Please correct these differences...

Thank you for the remark. The differences have been corrected.

*In order to measure temperature, wind speed and wind direction at different heights, you use the data of a meteorological mast located a few kilometers away from the wind turbine.*

– Since you have access to SCADA recordings, have you compared the wind speed and direction on the wind turbine hub with the ones measured by the anemometer on the meteo mast. It would allow you to check the spatial homogeneity of atmospheric quantities in this specific wind farm.

Thank you for this suggestion. A comparison of the SCADA recordings with the meteorological mast data has been included in the discussion (Sec. 4.3, l. 475-504) This will be further addressed in the discussion section below.

– What is the resolution of the temperature sensors you use? If this is 1 Hz as for the wind speed (line 170) it is likely to be insufficient to characterize turbulent fluctuations (see subsection below on atmospheric turbulence).

The available temperature data are averaged over ten minutes. In addition to the mean values, the standard deviations are also known. Unfortunately, the resolution of the temperature sensors is not known. The atmospheric turbulence is further addressed in the section below (see 1.3).

Page 15, you mention that background noise can change in the course of the measurement campaign depending on the vegetation. What is the duration of the measurement campaign? Could you add the date and hour of each case in Table 3? This would also be a valuable information regarding the change in ground impedance (see 1.4)).

The details about the time of the measurement campaign and the individual cases have been added to the paper (p.11, Table 3)

You calculate the effective sound speed in Equation (8) using  $c_{eff} = c_0 - |u| \cos \gamma$ . If  $\gamma$  is the angle between the wind direction and the sound propagation direction, with  $\gamma = 0$  corresponding to downwind conditions, the expression should be  $c_{eff} = c_0 + |u| \cos \gamma$ .

Thank you for that comment. This is indeed unclear in the paper. In Figure 4(a), the sound propagation direction was plotted instead of the wind direction. The sound propagation is opposite to the wind direction. In terms of wind conditions, 180° indicates downwind and 0° is upwind. Figure 4(a) has been corrected. In addition, the horizontal wind speed  $u_{comp}$  has been explained in more detail. With 180° as the downwind situation, Eq. (7) and (8)  $c_{eff} = c_0 - |u| \cos \gamma$  applies (l. 180-188).

You explain line 234 that a sampling rate of 5 Hz is used within the 1/3 octave bands. Is it true only for the measured data? Do you calculate the numerical SPL only at the center frequencies of the 1/3 octave band or you consider several frequencies per 1/3 octave band?

The measured acoustical time signals were recorded at a high resolution of 51 kHz. In addition, the sound pressure level meter stored 1/3 octave bands averaged over one second. The simulations were performed for frequencies from 70 to 2240 in 5 Hz steps. Several frequencies per 1/3 octave band are considered.

In Figure 4(a) the values of wind speed cannot be read. Also, why don't you plot values at heights closer to the hub (such as 100 m for wind speed and 95 m for wind direction)? They would be more representative of what the average inflow impinging on the wind turbine.

Figure 4(a) has been corrected. Originally, wind direction and wind speed were shown at mean height. It was thought that this data would most closely represent the general propagation conditions. In fact, it is also reasonable to plot the data from heights closer to the nacelle. This data is now presented in the paper.

*Figure 5 does not provide any additional information compared to Figure 4(b) and can be removed. The values of RMSE can simply be given in the text.*

We agree with this and have updated this part.

**1.3 Characterization of atmospheric turbulence**

To characterize the turbulent fields of temperature and wind speed, you use a Gaussian correlation function with a correlation length of 1m and a variance of index of refraction fluctuations  $\mu^2$ . This variance is related to the variance of the temperature and wind speed turbulent fluctuations, as given by Equation (6) of your paper.

To study the scattering of acoustic waves by turbulent structures, it is however necessary to measure temperature and wind speed fluctuations with a sampling frequency of at least 20 Hz (with an ultrasonic anemometer for instance). Indeed, the structures of interest are generally located in the inertial subrange (see Figures 1 and 3 of Wilson et al. [3]). If you use a sampling frequency of 1 Hz, you will only capture turbulent structures in the energy subrange, as can be seen for instance in Figure 8 of Daigle et al. [4].

*Figure 8 of Daigle et al. [4] also clearly shows that the Gaussian model is not adapted to capture the spectral slope in the inertial subrange. At best, it can fit the spectrum over a small range of turbulence wavenumbers, as explained in Wilson et al. [3], but the von Kármán model is generally recommended.*

Thus, I disagree with your sentence lines 518-519, where you conclude that "the available meteorological data, i.e. the wind speed and temperature fluctuations, are sufficient for the determination of atmospheric turbulence". I think that the experimental estimates of  $\mu^2$  cannot be used to calculate the turbulence spectrum. The good agreement that you obtain in Figure 11 at 845m with turbulence above 1000 Hz may not be meaningful. Indeed the SPL values are very low in the shadow zone, and corresponding signal-to-noise ratio is very low, as can be seen in Figure 9. Thus the difference between the model predictions without turbulence and the measurements above 1000 Hz is likely not be due (at least partly) to background noise.

We agree, high resolutions of the temperature and wind speed measurements are needed to determine the fast fluctuations. Unfortunately, the sampling rate of the temperature sensors is not known. Only 10-minute averages and associated standard deviations are available. For the wind speeds, a sampling frequency of 1 Hz is given. It is indeed questionable whether the available measurement data can be used to characterise atmospheric turbulence. We agree that the turbulent structures in the inertial subrange are not captured with the sampling frequency of the measurements. Furthermore, the sound propagation model used in the paper applies the Gaussian correlation function. As also explained in the review, the von Kármán model is recommended.

Due to these limitations, we decided to exclude atmospheric turbulence from the paper. Consequently, turbulence is not considered in the simulations with strong upwind conditions. With regard to Figure 9, the low signal-to-noise ratio has been addressed (l. 401-402).

**1.4 Ground effect**

You show clearly in Figure 15 that the ow resistivity used to calculate the ground impedance in the Delany-Bazley-Miki model has a significant impact on the propagation loss. From Figure 10 and Figure A.1, it appears that the modeled and measured third octave band spectra are in good agreement at 535 m, but not at 845 m, with differences greater than 5 dB between 250 Hz and 500 Hz approximately. The main reason for the discrepancies between model predictions and measurements is likely to be the ground effect, as refraction effects cannot explain the differences (small vertical sound speed gradient for cases 3/4 and cases 5/6). You mention that ground properties may have changed between the various cases lines 389-394, due for instance to the soil humidity. It has indeed been shown in the literature that the flow resistivity can vary drastically if the soil is dry or wet. In Figure 15 you have tested various flow resistivities at 535 m. Could you perform the same tests at 845m where the agreement is less good, to test if a change in flow resisitivity could explain the differences between model predictions and measurements?

Also, you show in Figure 14 that model predictions with a single point source yield strong interference minima and maxima, that are smoothed out when you consider your wind turbine model with 3 point sources, as shown by various researchers [5, 6]. However at 845m the interference minima and maxima associated with ground effect seem too strong in the model predictions compared to the measurements. This could mean that 3 point sources is not enough to calculate the propagation loss. Could you test with a greater number of point sources distributed over the rotor height?

Thank you for the detailed comment on the ground effect. As recommended, different flow resistances were investigated at a distance of 845 m from the turbine (Sec. 4.2, l. 452-474). Moreover, sound sources with different numbers of point sources (1, 3 and 10) were tested at the same distance (Sec. 4.1, l. 430-450). In view of the discrepancies between modelled and measured data at long distances, both tests were included in the discussion and are addressed below (see 1.6).

**1.5 Section 3.2.2 on total SPL**

In section 3.2.2, you discuss in length the agreement between measured and modelled propagation losses averaged over all frequencies. It seems to me that this is a bad indicator to check the validity of the model. For instance for case 4 at 845 m, you obtain a small difference of 0.15 dB, while the spectra in Figure 10 differ significantly. What happens is that the overestimation at low frequencies is compensated by the underestimation at high frequencies. This is even more problematic when you write lines 441-442: "Being 0.92 dB, the averaged absolute difference of the losses is also below 1 dB, meaning that the model generally predicts the propagation losses well also at longer distances." This averaging makes no sense to me! If you want to quantify the validity of your model predictions using a single number, my suggestion would be to calculate the mean over frequencies of the absolute value of the differences (or something similar)

That is right. Thank you for the good suggestion to calculate the average of the absolute values of the difference over frequency. It has been included in the paper (Sec. 3.3, l. 415-423). We think it is still good to compare the measured and modelled sound propagation loss in overall sound pressure levels (we have changed from total to overall sound pressure level). Even if it does not give information about validity, it is an important parameter.

**1.6 Section 3.3.1 on model assumptions and limitations**

In the review, it is recommended to focus on the main issues: 1. Source model 2. Ground properties 3. Atmospheric turbulence 4. Homogeneity of atmospheric parameters.

Atmospheric turbulence has been excluded from the paper. Hence, the discussion includes the source model, the ground effect and the homogeneity of atmospheric parameters.

1. Source model (Sec. 4.1, l. 430-450)

With regard to the source model, simulations were performed for varying numbers of point sources (1, 3, and 10). The ten point sources were distributed evenly over the rotor diameter. The difference between the simulations with one and three point sources has already been explained in various publications. The results with three and ten point sources show almost no difference. The validation results are not improved by simply increasing the number of point sources. However, it cannot be excluded that the discrepancies in the 1/3 octave spectrum are at least partly due to the simplified source modelling.

**2. Ground properties (Sec. 4.2, l. 452-474)**

In terms of ground properties, simulations were carried out with different values of flow resistivity. As the ground effect increases with distance, the results at a distance of 845 m are shown in the paper. For small values of flow resistivity, the peak becomes broader but is also shifted to lower frequencies. The validation results are not improved by changing the flow resistivity. The model assumes a constant flow resistivity along the propagation path. It is also possible that the ground conditions vary along the propagation path.

3. Homogeneity of atmospheric parameters (Sec. 4.3, l. 475-503)

The sound speed profile was determined using data from a 100m mast located 2 km northwest of the turbine under study. In order to assess the homogeneity of the atmospheric parameters, a comparison of the SCADA data with the 100m mast data was added to the paper. In addition, the sound speed was calculated from the SCADA data and plotted together with the sound speed profile determined from the mast data. In this way, the effect of differences between the wind direction measurements and the wind speed measurements could have been analysed. In most cases, small deviations were found between the calculated value (wind turbine) and the profile (mast). In two cases a deviation of 1 m/s was observed due to differences in the measured wind direction or wind speed. In these cases, the profile of the effective sound speed could change slightly if the SCADA data is included. However, the impact on the validation results is expected to be negligible. We believe that the use of the 100 m met mast data to determine the atmospheric conditions on the propagation path is valid.

**2. Technical corrections**

Many thanks for the technical corrections. They have all been included in the manuscript.

**Reviewer 3: (marked orange in the manuscript)**

Thank you very much for the inspiring review. We appreciate it very much. The impact of averaging time periods on validation results has been included in the paper. It is a very good addition regarding the measurement data and improves the quality of the paper.

The short term variability of meteorological parameters (e.g. wind speed, wind direction, that are part of low frequency turbulent motion) and heterogeneous ground properties is difficult to handle. This may lead to high variance in measured sound data and is difficult to compare to 'regular' model data. This variability is not discussed in detail. E.G. how it was determined the absence of a low level jet by mast measurements to 100m height? How the averaging of meteo and accoustic data affect the comparison of data? Which time range would be recommended (1min, 5 min, 10 min)?

Thank you for your comment and suggestion. It is indeed difficult to consider the short term variability of meteorological parameters and the heterogeneous ground properties. The absence of low-level jets was determined using the measurement data from the 100 m mast. Accordingly, only wind speeds between 27 and 100 m height were considered. It was verified that the wind speed increases continuously with height. A limitation is that low level jets above 100 m are not covered. (l. 220-221)

By averaging over a five minute period, we assume that the short-term changes in the meteorological parameters are averaged out and have no effect on the averaged sound pressure level. An analysis of the impact of parameter averaging and its effect on the validation results has been added to the manuscript (Sec. 4.4, l. 505-535).

In the case analysed, the wind direction does not depend significantly on the averaging time. Deviations in the wind profile are observed with an averaging time of one minute. The profile does not show the usual logarithmic curve. For the sound propagation losses, deviations are observed in the lower and upper frequencies with averaging times of one and ten minutes. For a few frequencies, the averaging time affects the validation results.

When using the data for validation purposes, we recommend an averaging time of three to five minutes. If the averaging time is shorter, the short term variability of meteorological parameters may affect the sound pressure levels. In addition, the measurement mast is about 2 km away from the propagation path, so the short-term changes cannot be transferred to the atmospheric conditions on the sound propagation path. An averaging time of ten minutes is not recommended due to the acoustic evaluation. To ensure that the wind turbine noise is dominant, the time signals should be listened to.

Here, the surprisingly good agreement between the model and the measurement gives rise to doubts about the selection of the cases examined. Normally, comparisons of measurement results with very similar conditions often show higher differences. Nevertheless, the results of this paper should be regarded as a first step how to proceed, about input parameters and comparison between measurements and model results.

The cases with similar environmental conditions were recorded within max. 1.5 hours. Thus no large differences in ground properties are expected. We think that examples with similar meteorological conditions, but recorded on different days, will show larger differences. For an exact comparison, the environmental conditions, e.g. also the ground properties, should be detected in more detail. In the script, the times of the validation cases have been added and commented (p.11, Table 3).

It is still unclear how a wind turbine should be represented in a sound propagation model. The method chosen here with three alternative sources is one of many possibilities, but there is no evidence that this is the best possible.

This is correct. There are many methods for modelling a wind turbine source. The method used here is simplified. It does not consider precise information about the sound source and its radiation. However, this information is not available. The sound source modelling is further addressed in the paper (see also Reviewer 2) (Sec. 4.1, l. 430-450). We hope that the measured data will be used by other researchers working with a more detailed source model. Such results would be very interesting.

The wide range of literature is presented, what may help any reader to advance how to proceed with measurements and comparison to model efforts.

The literature has been updated. One paper, which was published during the review process, has been added.

**Literature**

Nyborg, C. M., Fischer, A., Thysell, E., Feng, J, Sndergaard, L. S., Srensen, T, Hansen, T. R., Hansen, K. S., and Bertagnolio, F.: Propagation of wind turbine noise: measurements and model evaluation, Journal of Physics: Conference Series, Volume 2265, 2022.

**References from the review**

[1] S. Oerlemans, P. Sijtsma and B. Mndez Lpez, Effect of wind shear on amplitude modulation of wind turbine noise, Journal of Sound and Vibration, vol. 299, pp. 869-883, 2007.

[2] S. Oerlemans, Effect of wind shear on amplitude modulation of wind turbine noise, International Journal of Aeroacoustics, vol. 14, pp. 715-728, 2015.

[3] D.K. Wilson, J.G. Brasseur, and K.E. Gilbert, Acoustic scattering and the spectrum of atmospheric turbulence, Journal of the Acoustical Society of America, vol. 105, pp.30-34,1999.

[4] G. A. Daigle, T. F. W. Embleton, and J. E. Piercy, Propagation of sound in the presence of gradients and turbulence near the ground, Journal of the Acoustical Society of America, vol. 79, pp. 613-627, 1986.

[5] K. Heutschi, R. Pieren, M. Müller, M. Manyoky, U.W. Hayek, K. Eggenschwiler, Auralization of Wind Turbine Noise: Propagation Filtering and Vegetation Noise Synthesis, Acta Acustica united with Acustica, vol. 100, pp. 13-24, 2014.

[6] B. Cottè, Extended source models for wind turbine noise propagation, Journal of the Acoustical Society of America, vol. 145, pp. 1363-1371, 2019